# The membrane domains of mammalian adenylyl cyclases are lipid receptors

Marius Landau[1], Sherif Elsabbagh[1], Harald Gross[1], Adrian CD Fuchs[2], Anita CF Schultz[1], Joachim E Schultz[1]*

[1]Pharmazeutisches Institut der Universität Tübingen, Tübingen, Germany; [2]Max-Planck-Institut für Biologie, Tübingen, Germany

*For correspondence:
joachim.schultz@uni-tuebingen.de

Competing interest: The authors declare that no competing interests exist.

## eLife Assessment

This manuscript describes an **important** study of a new lipid-mediated regulation mechanism of adenylyl cyclases. The biochemical evidence provided is **convincing** and will trigger more research in this mechanism. This manuscript will be of interest to all scientists working on lipid regulation and adenylyl cyclases.

**Abstract** The biosynthesis of cyclic 3′,5′-adenosine monophosphate (cAMP) by mammalian membrane-bound adenylyl cyclases (mACs) is predominantly regulated by G-protein-coupled receptors (GPCRs). Up to now the two hexahelical transmembrane domains of mACs were considered to fix the enzyme to membranes. Here, we show that the transmembrane domains serve in addition as signal receptors and transmitters of lipid signals that control Gsα-stimulated mAC activities. We identify aliphatic fatty acids and anandamide as receptor ligands of mAC isoforms 1–7 and 9. The ligands enhance (mAC isoforms 2, 3, 7, and 9) or attenuate (isoforms 1, 4, 5, and 6) Gsα-stimulated mAC activities in vitro and in vivo. Substitution of the stimulatory membrane receptor of mAC3 by the inhibitory receptor of mAC5 results in a ligand inhibited mAC5–mAC3 chimera. Thus, we discovered a new class of membrane receptors in which two signaling modalities are at a crossing, direct tonic lipid and indirect phasic GPCR–Gsα signaling regulating the biosynthesis of cAMP.

## Introduction

The structure of the first second messenger, cyclic 3′,5′-adenosine monophosphate (cAMP), was reported in 1958 by Sutherland and Rall. It was generated by incubation of a liver extract with the first messengers adrenaline or glucagon (*Sutherland and Rall, 1958*). Since then, cAMP has been demonstrated to be an almost universal second messenger used to translate various extracellular stimuli into a uniform intracellular chemical signal (*Dessauer et al., 2017*; *Ostrom et al., 2022*). The biosynthetic enzymes for cAMP from ATP, adenylyl cyclases (ACs), have been biochemically investigated in bacteria and eukaryotic cells (*Khandelwal and Hamilton, 1971*; *Linder and Schultz, 2003*; *Linder and Schultz, 2008*). Finally in 1989, the first mammalian AC was sequenced (*Krupinski et al., 1989*). The protein displayed two catalytic domains (C1 and C2) which are highly similar in all isoforms. The two hexahelical membrane anchors (TM1 and TM2) are dissimilar and isoform specifically conserved (*Bassler et al., 2018*). The latter were proposed to possess a channel or transporter-like function, properties, which were never confirmed (*Krupinski et al., 1989*). The domain architecture of mammalian ACs, TM1–C1–TM2–C2, clearly indicated a pseudoheterodimeric protein composed of two concatenated monomeric bacterial precursor proteins (*Guo et al., 2001*). To date, sequencing has identified nine mammalian membrane-delimited AC isoforms (mACs) with identical domain architectures (*Dessauer et al., 2017*). In all nine isoforms the catalytic domains are conserved and share extensive sequence

and structural similarities which indicate a similar biosynthetic mechanism (*Linder and Schultz, 2003*; *Linder and Schultz, 2008*; *Tesmer et al., 1997*; *Sinha and Sprang, 2006*). On the other hand, the associated two membrane domains differ substantially within each isoform and between all nine mAC isoforms (*Bassler et al., 2018*; *Schultz, 2022*; *Tang and Gilman, 1995*). Bioinformatic studies, however, revealed that the membrane domains, TM1 as well as TM2, are highly conserved in an isoform-specific manner for about 0.5 billion years of evolution (*Bassler et al., 2018*; *Schultz, 2022*).

Extensive studies on the regulation of the nine mAC isoforms revealed that the Gsα subunit of the trimeric G proteins activate cAMP formation. Gsα is released intracellularly upon stimulation of G-protein-coupled receptors (GPCRs), i.e., the receptor function for mAC regulation was assigned to the diversity of GPCRs, which presently are most prominent drug targets. In 1995 Tang and Gilman reported that Gsα regulation of mammalian mACs does not require the presence of the membrane anchors. A soluble C1–C2 dimer devoid of the membrane domains was fully activated by Gsα (*Tang and Gilman, 1995*). This reinforced the view that the mAC membrane anchors which comprise up to 40% of the protein were just that and otherwise functionally inert. The theoretical possibility of mACs to be regulated directly, bypassing the GPCRs, was dismissed (*Schuster et al., 2024*).

We were intrigued by the exceptional evolutionary conservation of the mAC membrane anchors for 0.5 billion years (*Bassler et al., 2018*; *Schultz, 2022*). In addition, we identified a cyclase-transducing element that connects the TM1 and TM2 domains to the attached C1 or C2 catalytic domains. These transducer elements are similarly conserved in a strictly isoform-specific manner (*Schultz, 2022*; *Ziegler, 2017*). Furthermore, cryo-EM structures of mAC holoenzymes clearly revealed that the two membrane domains, TM1 and TM2, collapse into a tight dodecahelical complex resembling membrane receptors (*Qi et al., 2019*; *Qi et al., 2022*; *Gu et al., 2001*). Lastly, we created a chimeric model involving the hexahelical quorum-sensing receptor from *Vibrio cholerae* which has a known aliphatic lipid ligand and the mAC 2 isoform. We observed that the ligand directly affected, i.e., attenuated the Gsα activation of mAC2 (*Seth et al., 2020*). As a proof-of-concept this demonstrated that the cytosolic catalytic AC dimer serves as a receiver for extracellular signals transmitted through the dyad-related membrane anchor. Accordingly, we proposed a general model of mAC regulation in which the extent of the indirect mAC activation via the GPCR/Gsα axis is under direct ligand control via the mAC membrane anchor (*Seth et al., 2020*).

Here, we report the results of a rigorous search for potential ligands of the mAC membrane domains. We identified aliphatic lipids as ligands for mAC isoforms 1–7 and 9. Isoform dependently, the ligands either attenuate or enhance Gsα activation in vitro and in vivo demonstrating a receptor function of the mAC proteins. The receptor properties are transferable as demonstrated by interchanging the membrane anchors between mAC3 and 5. Thus, the results define a new class of membrane receptors and establish a completely new level of regulation of cAMP biosynthesis in mammals in which tonic and phasic signaling processes intersect in a central signaling system, which is the target of frequently used drugs.

## Results

### Oleic acid enhances Gsα-stimulated mAC3, but not mAC5 activity

In earlier experiments, we demonstrated regulation of the mycobacterial AC Rv2212 by lipids (*Abdel Motaal et al., 2006*), the presence of oleic acid in the mycobacterial AC Rv1264 structure (*Findeisen et al., 2007*), and regulation of a chimera consisting of the quorum-sensing receptor from *Vibrio* and the mAC2 catalytic dimer by the aliphatic lipid 3-hydroxytridecan-4-one (*Beltz et al., 2016*). Therefore, we searched for lipids as ligands. Here, we used bovine lung as a starting material because lipids are important for lung development and function (*Änggård and Samuelsson, 1965*; *Dautel et al., 2017*). Lipids were extracted from a cleared lung homogenate, acidified to pH 1, with dichloromethane/methanol (2:1). The dried organic phase was chromatographed on silica gel (employing vacuum liquid chromatography; Si-VLC) and fractions A to Q were assayed (*Figure 1—figure supplement 1*). Fraction E enhanced mAC3 activity stimulated by 300 nM Gsα fourfold, whereas mAC5 activity was unaffected (*Figure 1A*). The non-maximal concentration of 300 nM Gsα was used because it enabled us to observe stimulatory as well as inhibitory effects.

Fraction E was further separated by reversed-phase high-performance liquid chromatography (RP-HPLC) into five subfractions (E1–E5; *Figure 1—figure supplement 2*). The mAC3 enhancing

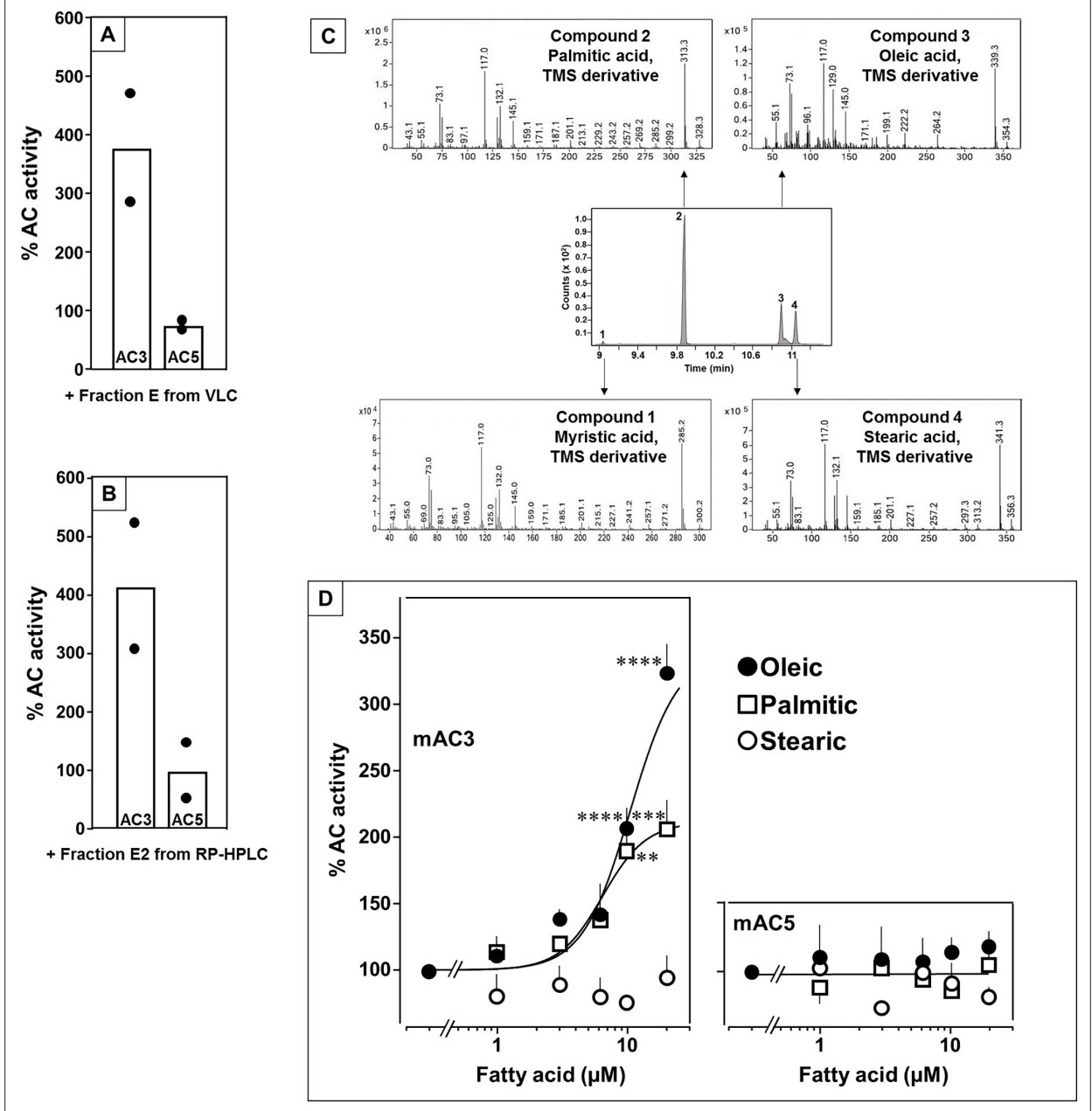

**Figure 1.** Identification of mAC3 activating fatty acids. Effect of 1 µg/assay of fractions E from vacuum liquid chromatography (**A**) and E2 from reversed-phase high-performance liquid chromatography (RP-HPLC) (**B**) on 300 nM Gsα-stimulated mAC isoforms 3 and 5. Activities are shown as % compared to 300 nM Gsα stimulation (100%). $n = 2$, each with two technical replicates. Basal and Gsα activities of mAC3 in (**A**) were 0.01 and 0.07 and of mAC5 0.06 and 1.32 nmol cAMP•mg$^{-1}$•min$^{-1}$, respectively. In (**B**), basal and Gsα activities of mAC3 were 0.02 and 0.12 and of mAC5 0.09 and 1.1 nmol cAMP•mg$^{-1}$•min$^{-1}$, respectively. (**C**) Gas chromatography–mass spectrometry (GC–MS) chromatogram of fraction E2. Mass spectra of the fatty acids are shown. Fatty acids' identity was confirmed by comparing with corresponding standards (TMS: Trimethylsilyl). (**D**) Effect of fatty acids identified by GC–MS on 300 nM Gsα-stimulated mAC3 (left) and mAC5 (right). Basal and Gsα activities of mAC3 were 0.023 ± 0.02 and 0.17 ± 0.03 and of mAC5 0.08 ± 0.02 and 0.44 ± 0.09 nmol cAMP•mg$^{-1}$•min$^{-1}$, respectively. $n = 3$–23. EC$_{50}$ of palmitic and oleic acids for mAC3 were 6.4 and 10.4 µM, respectively. Data represent individual experiments (black dots in A and B) or mean ± SEM (D). One-sample $t$ tests were performed. Significances: **$p < 0.01$; ***$p < 0.001$; ****$p < 0.0001$.

The online version of this article includes the following source data and figure supplement(s) for figure 1:

**Figure supplement 1.** Si-VLC: silica-vacuum liquid chromatography, RP-HPLC: reversed-phase high-performance liquid chromatography, EA: ethyl acetate, PE: petroleum ether, MeOH: methanol.

**Figure supplement 2.** Reversed-phase high-performance liquid chromatography (RP-HPLC) chromatogram of fraction E.

*Figure 1 continued on next page*

*Figure 1 continued*

**Figure supplement 3.** NMR spectra of fraction E2 in $d_4$-MeOH.

**Figure supplement 4.** Time-dependent stimulation of mAC3 by oleic acid.

**Figure supplement 5.** Hanes–Woolf plot of mAC3 ± 20 µM oleic acid.

**Figure supplement 6.** Oleic acid has no stimulatory effect on the soluble catalytic dimer.

**Source data 1.** Including data used for generating *Figure 1A, B, D*.

**Figure supplement 4—source data 1.** Including data used for generating *Figure 1—figure supplement 4*.

**Figure supplement 5—source data 1.** Including data used for generating *Figure 1—figure supplement 5*.

**Figure supplement 6—source data 1.** Including data used for generating *Figure 1—figure supplement 6*.

constituents appeared in fraction E2. It enhanced Gsα-stimulated mAC3 fourfold but had no effect on mAC5 (*Figure 1B*). [1]H- and [13]C-NMR spectra of fraction E2 indicated the presence of aliphatic lipids (*Figure 1—figure supplement 3*). Subsequent GC/MS analysis identified palmitic, stearic, oleic, and myristic acid in E2 (*Figure 1C*). Concentration–response curves were established for these fatty acids with mAC3 and mAC5 stimulated by 300 nM Gsα (*Figure 1D*). 20 µM oleic acid enhanced Gsα-stimulated mAC3 activity threefold ($EC_{50}$ = 10.4 µM) and 20 µM palmitic acid twofold ($EC_{50}$ = 6.4 µM), while stearic or myristic acid had no significant effect. None of these fatty acids affected mAC5 activity (*Figure 1D*).

The action of oleic acid on mAC3 was linear for >25 min (*Figure 1—figure supplement 4*). The Km of mAC3 for ATP (335 µM) was unaffected. Vmax was increased from 0.62 to 1.23 nmol cAMP/mg/min (*Figure 1—figure supplement 5*). Oleic acid did not affect the activity of a soluble, Gsα-stimulated construct formerly used for generating a C1 and C2 catalytic dimer from mAC1 and 2, ruling out spurious detergent effects (*Tang and Gilman, 1995*; *Figure 1—figure supplement 6*). The effect of oleic acid was further evaluated by Gsα concentration–response curves of mAC3 and mAC5 in presence and absence of 20 µM oleic acid (*Figure 2*, left and center). For mAC3 the calculated $EC_{50}$ of Gsα

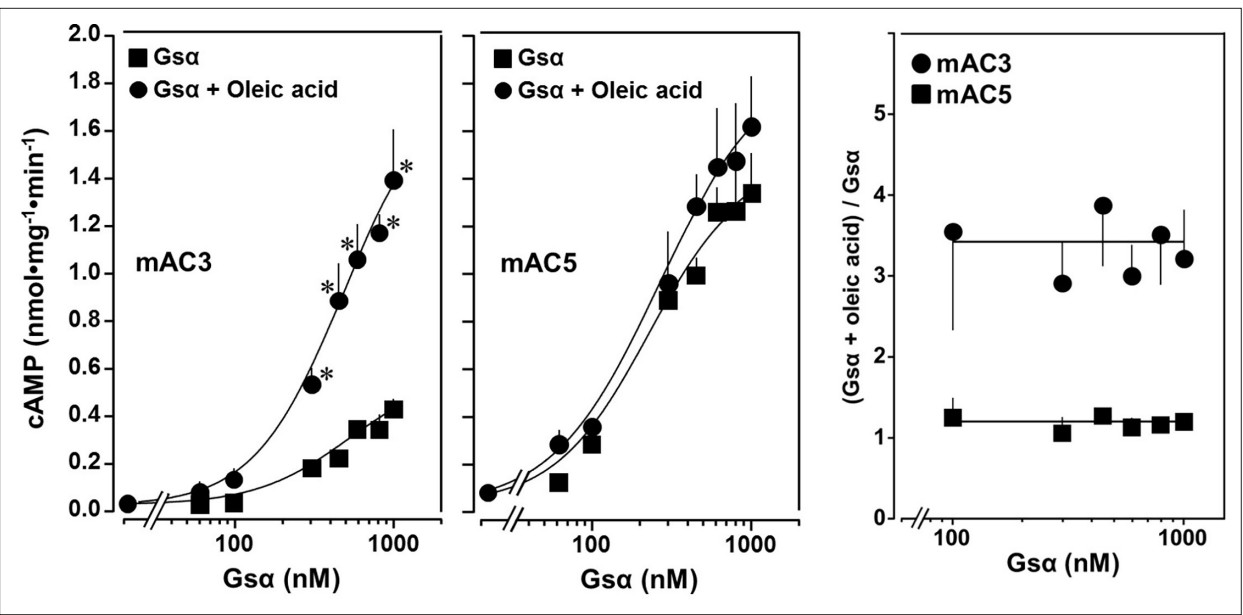

**Figure 2.** Concentration–response curves for Gsα and mAC3 and five activities in presence or absence of 20 µM oleic acid. 20 µM oleic acid enhances Gsα stimulation of mAC3 (left) but not of mAC5 (center). (Left) $EC_{50}$ of Gsα in the absence of oleic acid was 549 nM and in the presence of 20 µM oleic acid, it was 471 nM (not significant). mAC3 basal activity was 30 ± 24 pmol cAMP•mg$^{-1}$•min$^{-1}$. $n$ = 3, each with two technical replicates. (Center) The $EC_{50}$ of Gsα in the absence of oleic acid was 245 nM and in the presence of oleic acid, it was 277 nM (not significant). mAC5 basal activity was 84 ± 60 cAMP•mg$^{-1}$•min$^{-1}$. $n$ = 2, each with two technical replicates. (Right) (Gsα + oleic acid stimulation)/(Gsα stimulation) ratio of mAC3 and mAC5 from left and center ($n$ = 2–3). Data are mean ± SEM, paired $t$ test for left and center, and one-way ANOVA for right. Significances: *$p < 0.05$.

The online version of this article includes the following source data for figure 2:

**Source data 1.** Including data used for generating *Figure 2*, left, center, and right.

**Table 1.** List of lipids tested against mAC isoforms.

| |
|---|
| Lauric (dodecanoic) acid |
| Myristic (tetradecanoic) acid |
| Myristoleic ((9Z)-tetradecenoic) acid |
| Palmitic (hexadecanoic) acid |
| Palmitoleic ((9Z)-hexadecenoic) acid |
| Octadecane |
| 1,18-Octadecanedicarboxylic acid |
| Stearic (octadecanoic) acid |
| 9-Hydroxystearic acid |
| Oleic ((9Z)-octadecenoic) acid |
| Oleamide ((9Z)-octadecenamide) |
| Methyl oleate |
| 2-Oleoylglycerol |
| Triolein |
| Elaidic ((9E)-octadecenoic) acid |
| cis-Vaccenic ((11E)-octadecenoic) acid |
| Linoleic ((9Z,12Z)-octadecadienoic) acid |
| Linolenic ((9Z,12Z,15Z)-octadecatrienoic) acid |
| Arachidonic ((5Z,8Z,11Z,14Z)-eicosatetraenoic) acid |
| Eicosapentaenoic ((5Z,8Z,11Z,14Z,17Z)-eicosapentaenoic) acid |

in presence and absence of oleic acid were 549 and 471 nM, respectively (not significant). Over the Gsα concentration range tested with mAC3 the enhancement of cAMP formation by 20 μM oleic was uniformly about 3.4-fold (**Figure 2**, right). In the case of mAC5, Gsα stimulation was not enhanced by oleic acid (**Figure 2**, center and right).

To explore the ligand space, we tested 18 aliphatic $C_{12}$ to $C_{20}$ lipids (**Table 1**; structures in **Figure 3—figure supplement 1**). At 20 μM, elaidic, cis-vaccenic and linoleic acids were efficient enhancers of Gsα-stimulated mAC3 activity. Palmitic, palmitoleic, linolenic, eicosa-pentaenoic acids, and oleamide were less efficacious; other compounds were inactive (**Figure 3**). Notably, the saturated $C_{18}$ stearic acid was inactive here and throughout, albeit otherwise variations in chain length, and the number, location, and conformation of double bonds were tolerated to some extent, e.g., cis-vaccenic, linoleic, and linolenic acids. The relaxed ligand specificity was anticipated as aliphatic fatty acids are highly bendable and bind to a flexible dodecahelical protein dimer embedded in a fluid lipid membrane. The ligand space of mAC3 somewhat resembled the fuzzy and overlapping ligand specificities of the free fatty acid receptors 1 and 4 (**Kimura et al., 2020**; **Samovski et al., 2023**; **Grundmann et al., 2021**).

Next, the effect of oleic acid was probed in vivo using HEK293 cells permanently transfected with mAC3 (HEK-mAC3) or mAC5 (HEK-mAC5). Intracellular cAMP formation via Gsα was triggered via stimulation of the endogenous β-receptor with 2.5 μM of the β-agonist isoproterenol (concentration of isoproterenol is based on a respective concentration–response curve with HEK-mAC3 cell; see **Figure 4—figure supplement 1**). Addition of oleic acid enhanced cAMP formation in HEK-mAC3 1.5-fold (**Figure 4**). Stearic acid was inactive. Under identical conditions, cAMP formation in HEK-mAC5 cells was unaffected (**Figure 4**). The $EC_{50}$ of oleic acid in HEK293-mAC3 cells was 0.5 μM, i.e., the potency appeared to be increased compared to respective membrane preparations whereas the efficiency was reduced, possibly reflecting the regulatory interplay within the cell. To exclude experimental artifacts, transfection efficiencies were tested by PCR. mAC3 and mAC5 transfections were similar (**Figure 4—figure supplement 2**). Taken together, the results suggest that the enhancement of Gsα-stimulated mAC3 by oleic acid might be due to binding of oleic acid to or into an mAC3 membrane receptor (**Schultz, 2022**; **Beltz et al., 2016**).

## Oleic acid enhances Gsα-stimulated mAC 2, 7, and 9 activities

Next, we examined other AC isoforms with oleic acid as a ligand. 20 μM oleic acid significantly enhanced Gsα-stimulated activities of isoforms 2, 7, and 9, mAC1 was slightly attenuated, and isoforms 4, 5, 6, and 8 were unaffected (**Figure 5A**).

Concentration–response curves were carried out for mACs 2, 7, and 9 (**Figure 5B**). The $EC_{50}$ of oleic acid were 8.6, 6.7, and 7.8 μM, respectively, comparable to that determined for mAC3. Exploration of the ligand space for mACs 2, 7, and 9 with the panel of 18 aliphatic lipids uncovered more active lipids (**Figure 5C**). In the case of mAC2, 20 μM cis-vaccenic acid doubled cAMP formation ($EC_{50}$ 10.6 μM) while other compounds were inactive (**Figure 5C** and for additional concentration–response curves see **Figure 5—figure supplement 1**). For mAC7 the $EC_{50}$ of elaidic was 9.7 μM

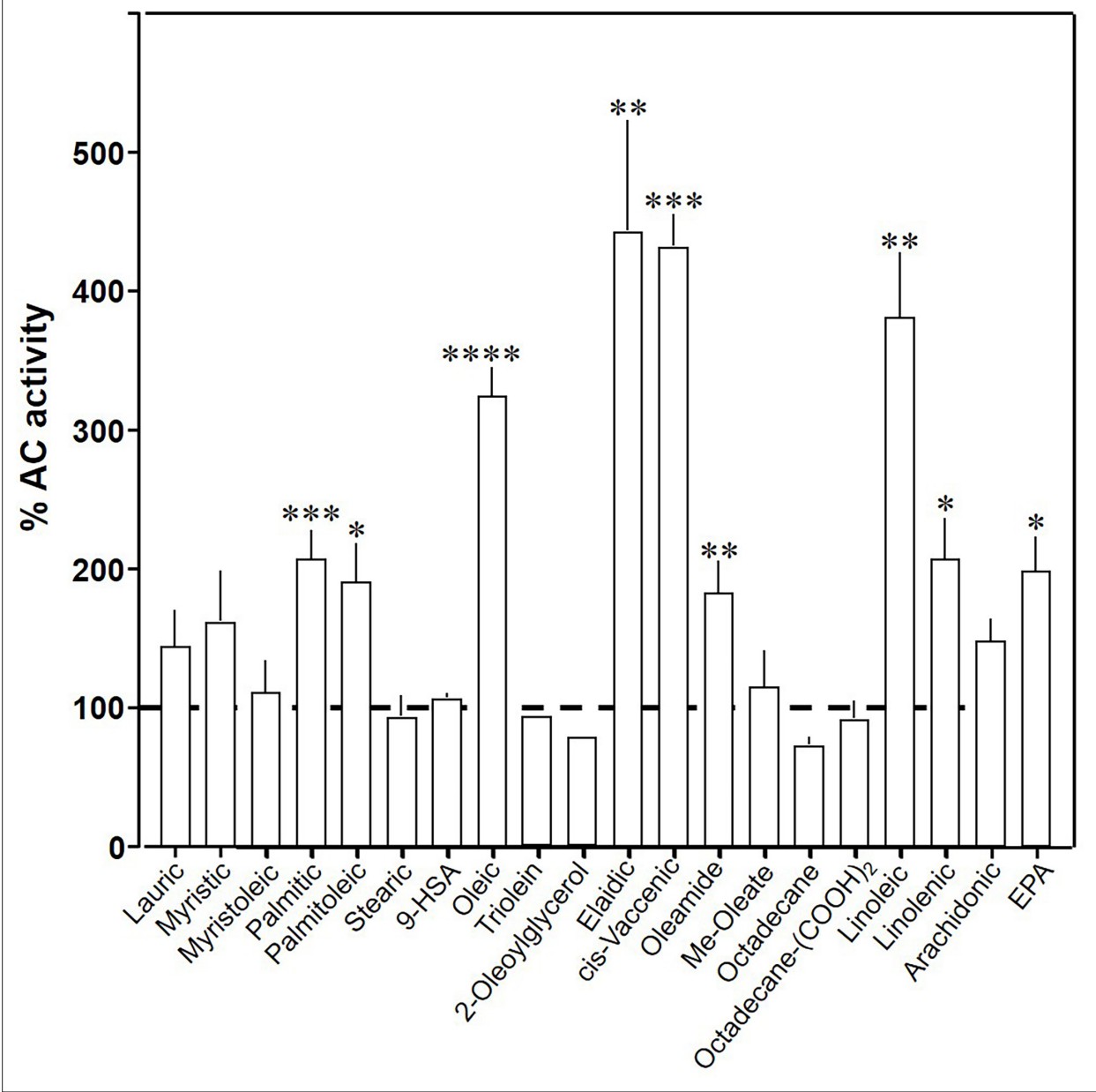

**Figure 3.** Effect of 20 µM lipids on 300 nM Gsα-stimulated mAC3. Basal and Gsα-stimulated activities were 0.02 ± 0.001 and 0.17 ± 0.01 nmol cAMP•mg$^{-1}$•min$^{-1}$, respectively. EPA: eicosapentaenoic acid; 9-HSA: 9-hydroxystearic acid. Data are mean ± SEM, $n$ = 2–23. One-sample $t$ tests, Significances: *p < 0.05; **p < 0.01; ***p < 0.001; ****p < 0.0001.

The online version of this article includes the following source data and figure supplement(s) for figure 3:

**Source data 1.** Including data used for generating *Figure 3*.

**Figure supplement 1.** Structures of the tested aliphatic lipids listed in *Table 1*.

(concentration–response curve see *Figure 5—figure supplement 2*). The range of potential ligands for mAC9 was more comprehensive: three- to fourfold enhancement was observed with 20 µM palmitoleic, oleic, elaidic, *cis*-vaccenic, and linoleic acid. With 20 µM myristic, palmitic, palmitoleic, linolenic, and arachidonic acid 1.5- to 2-fold enhancements were observed (*Figure 5C* and for concentration–response curves see *Figure 5—figure supplements 3 and 4*).

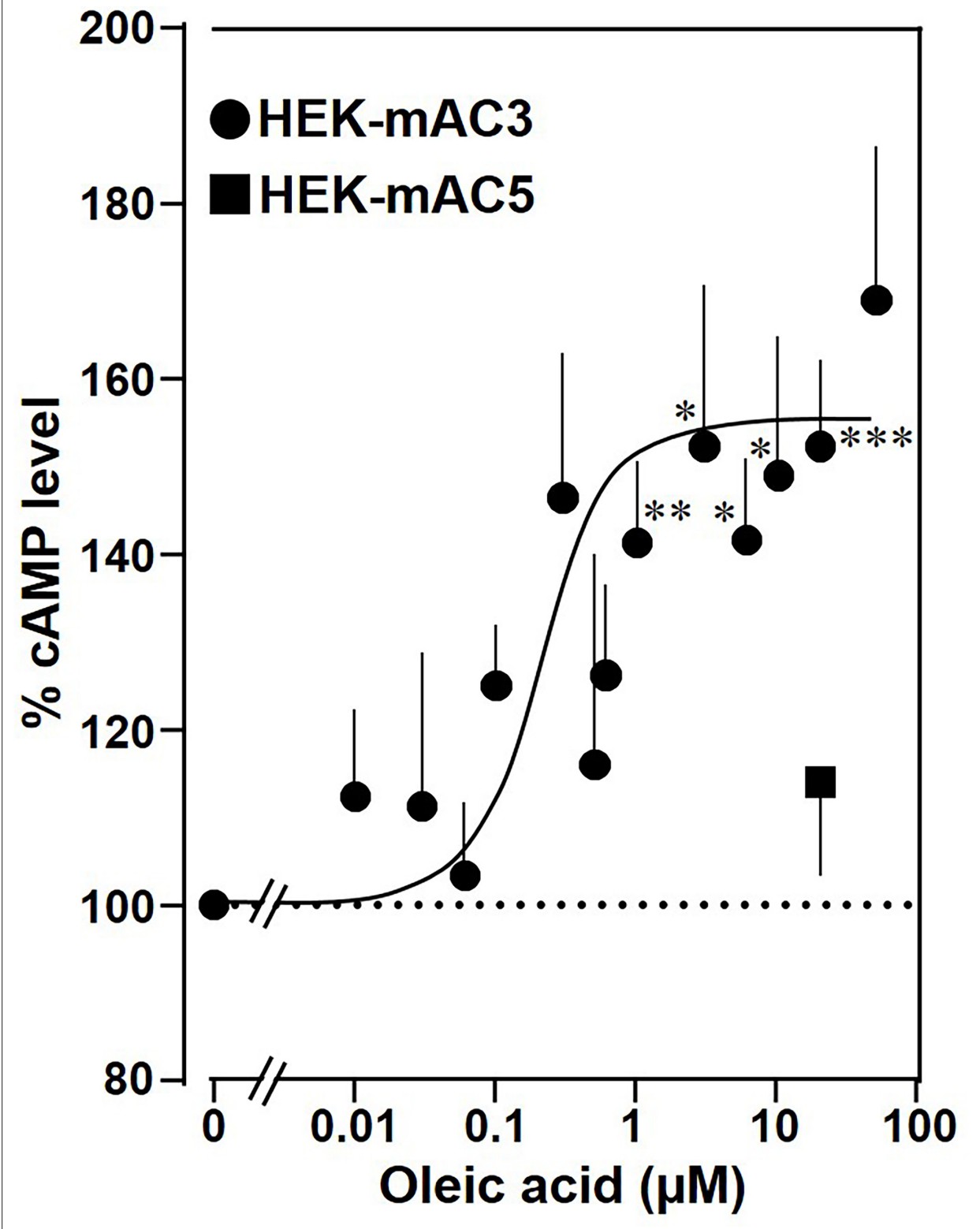

**Figure 4.** Oleic acid enhances cAMP formation in mAC3-transfected HEK293 cells. Effect of oleic acid on HEK293 cells permanently transfected with mACs 3 and 5 stimulated by 2.5 µM isoproterenol (set as 100%). Basal and isoproterenol-stimulated cAMP levels of HEK-mAC3 were 0.02 ± 0.006 and 1.35 ± 0.24 and of HEK-mAC5 2.13 ± 0.69 and 2.60 ± 0.88 pmol cAMP/10,000 cells, respectively. $n$ = 3–9, carried out in technical triplicates. Data are mean ± SEM. One-sample $t$ tests. Significances: *$p < 0.05$; **$p < 0.01$; ***$p < 0.001$.

The online version of this article includes the following source data and figure supplement(s) for figure 4:

*Figure 4 continued on next page*

*Figure 4 continued*

**Figure supplement 1.** Effect of isoproterenol on cAMP formation in HEK293 cells transfected with mAC3.

**Figure supplement 2.** Agarose gels of PCR products from HEK293 cells permanently transfected with mAC1–9.

**Source data 1.** Including data used for generating *Figure 4*.

**Figure supplement 1—source data 1.** Including data used for generating *Figure 4—figure supplement 1*.

**Figure supplement 2—source data 1.** PDF file containing original gels for *Figure 4—figure supplement 2*, indicating the relevant bands.

**Figure supplement 2—source data 2.** Original files containing gels for *Figure 4—figure supplement 2*.

## Arachidonic acid and anandamide inhibit Gsα-stimulated activities of mAC1, 4, 5, and 6

Testing the panel of lipids at 20 μM with mAC isoforms 1, 4, 5, 6, and 8 we found that isoforms 1 and 4 were significantly attenuated by arachidonic acid, and somewhat less by palmitoleic acid. Other lipids had no effect (for bar plots and dose–response curves see *Figure 6—figure supplements 1–3*). Of note, eicosa-pentaenoic acid which resembles arachidonic acid but for an additional *cis*-$\Delta^{-17}$ double bond had no effect on mAC activities (*Figure 6—figure supplements 1 and 2*). Concentration–response curves for arachidonic acid with 300 nM Gsα-stimulated mAC1 and 4 yielded $IC_{50}$ of 23 and 36 μM, respectively, i.e., about twofold higher compared to the $EC_{50}$ of enhancing ligands (*Figure 6A*).

Next, we examined whether arachidonic acid attenuates mAC1 and 4 in intact HEK 293 cells. Surprisingly, cAMP formation in HEK-mAC1 cells stimulated by 10 μM isoproterenol was attenuated by arachidonic acid with high potency ($IC_{50}$ = 250 pM), i.e., with higher potency compared to data with membranes prepared from the same cell line. In contrast, mAC4 activity examined under identical conditions was not attenuated (*Figure 6B*). Currently, we are unable to rationalize these discrepancies. Possibly, mAC4 has another, more specific lipid ligand which is needed in in vivo. In general, the enhancing and attenuating effects bolster the hypothesis of specific receptor-ligand interactions and divergent intrinsic activities for different ligands. Of note, it was reported that arachidonic acid at concentrations up to 1 mM inhibits AC activity in brain membrane fractions and that essential fatty acid deficiency effects AC activity in rat heart (*Nakamura et al., 2001*; *Alam et al., 1995*).

At this point we were lacking ligands for mACs 5, 6, and 8 (*Figure 7—figure supplements 1 and 2* and Figure 8). Possibly, the negative charge of the fatty acid headgroups might impair receptor interactions. A neutral lipid neurotransmitter closely related to arachidonic acid is arachidonoylethanolamide (anandamide) (*Mock et al., 2023*). Indeed, anandamide attenuated 300 nM Gsα stimulation of mAC5 and 6 with $IC_{50}$ of 42 and 23 μM, respectively, i.e., comparable to the effect of arachidonic acid on mACs 1 and 4, and distinctly less potently than the ligands for mAC 2, 3, 7, and 9 (*Figure 7A*). mACs 5 and 6 may thus represent new targets for anandamide which is part of a widespread neuromodulatory system (*Lu and Mackie, 2016*). The concentrations of arachidonic acid and anandamide required may be achieved in vivo by local biosynthesis and degradation. An interfacial membrane-embedded phosphodiesterase cleaves the phosphodiester bond of the membrane lipid *N*-arachidonoyl-ethanolamine-glycerophosphate releasing anandamide into the extracellular space (*Mock et al., 2023*; *Simon and Cravatt, 2008*; *Liu et al., 2006*). The lipophilicity and lack of charge should enable it to diffuse readily. Whether the mACs and this biosynthetic phosphodiesterase colocalize or associate with its target mACs is unknown. Degradation of anandamide is by a membrane-bound amidase, generating arachidonic acid and ethanolamine (*McKinney and Cravatt, 2005*). Therefore, we examined whether anandamide at higher concentrations might also affect mAC1 and 4. In fact, anandamide significantly attenuated Gsα-stimulated mAC1, but distinctly not mAC4 (*Figure 7B*). The $IC_{50}$ of anandamide for mAC1 was 29 μM. We also tested whether anandamide attenuated cAMP formation in vivo using HEK-mAC5 cells primed by 2.5 μM isoproterenol (*Figure 7C*). 100 μM anandamide attenuated cAMP formation by only 23% in HEK293-mAC5 cells, the effect was significant (p < 0.01). At this point, we were unable to identify a ligand for mAC8, presumably another lipid (*Figure 8*).

## Receptor properties are exchangeable between mAC isoforms 3 and 5

To unequivocally validate specific mAC–ligand–receptor interactions and regulation we generated a chimera in which the enhancing membrane domains of mAC3, i.e., mAC3-TM1 and TM2, were

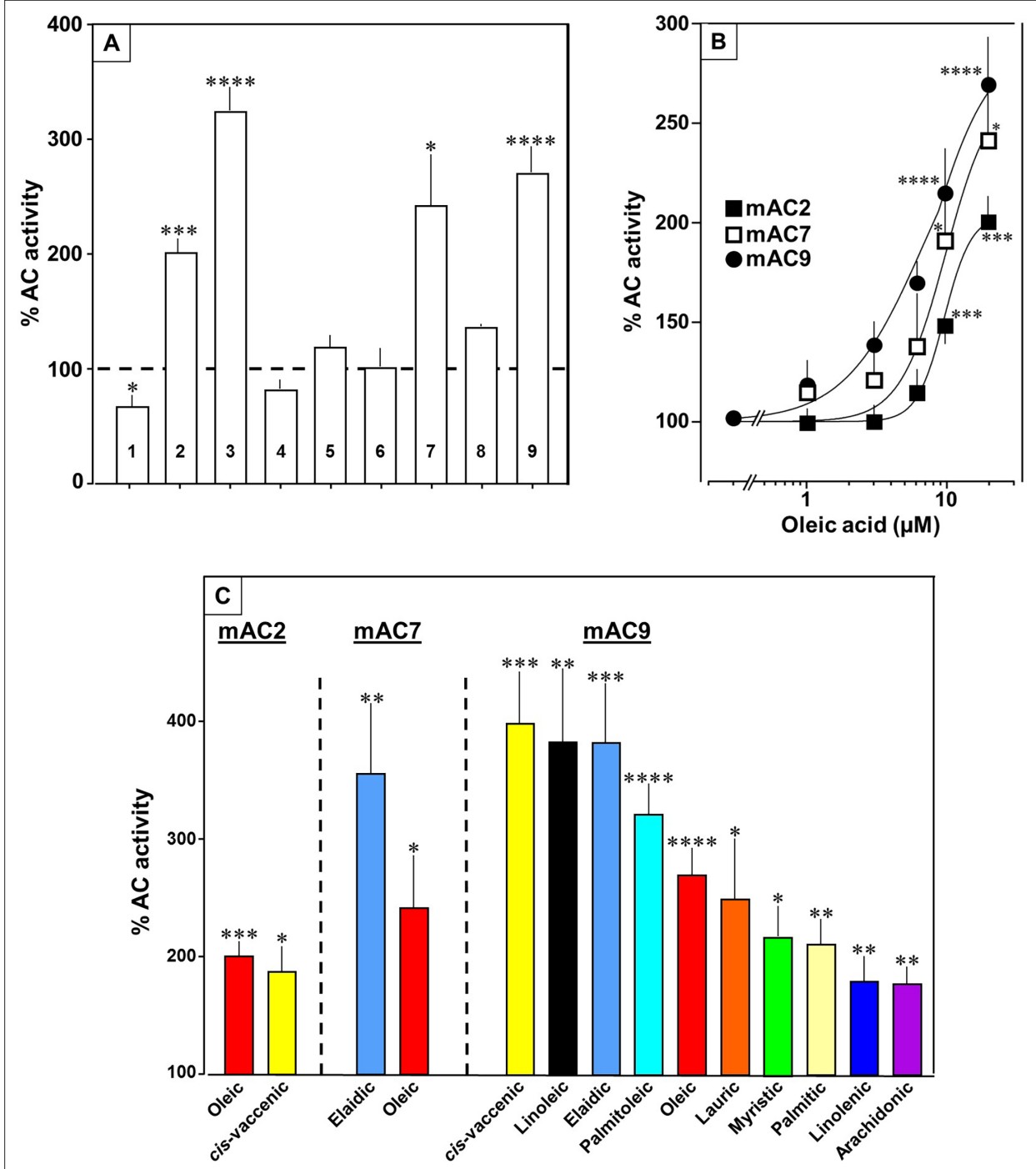

**Figure 5.** Fatty acids enhance mAC isoforms 2, 7, and 9 activities. (**A**) Effect of 20 µM oleic acid on 300 nM Gsα-stimulated mAC activities normalized to 100%. Basal and Gsα-stimulated activities for each isoform are in *Figure 5—figure supplement 1*. n = 2–23. (**B**) Oleic acid activates mACs 2, 7, and 9 stimulated by 300 nM Gsα. n = 7–23. (**C**) Fatty acids activating mACs 2, 7, and 9 at 20 µM. For basal and Gsα-stimulated activities, see *Figure 5—figure supplements 1–4*. n = 5–15. Identical colors indicate identical compounds. Data are mean ± SEM. One-sample *t* tests were performed. Significances: *p < 0.05; **p < 0.01; ***p < 0.001; ****p < 0.0001.

The online version of this article includes the following source data and figure supplement(s) for figure 5:

**Source data 1.** Including data used for generating *Figure 5A–C*.

**Figure supplement 1.** Effect of lipids on 300 nM Gsα-stimulated mAC2.

**Figure supplement 1—source data 1.** Including basal and Gsα-stimulated activities of mAC1–mAC9.

*Figure 5 continued on next page*

*Figure 5 continued*

**Figure supplement 1—source data 2.** Including data used for generating *Figure 5—figure supplement 1*, left and right.

**Figure supplement 2.** Effect of lipids on 300 nM Gsα-stimulated mAC7.

**Figure supplement 2—source data 1.** Including data used for generating *Figure 5—figure supplement 2*, left and right.

**Figure supplement 3.** Effect of lipids on 300 nM Gsα-stimulated mAC9.

**Figure supplement 3—source data 1.** Including data used for generating *Figure 5—figure supplement 3*, left and right.

**Figure supplement 4.** Concentration–response curves of fatty acids activating Gsα-stimulated mAC9.

**Figure supplement 4—source data 1.** Including data used for generating *Figure 5—figure supplement 4*, left and right.

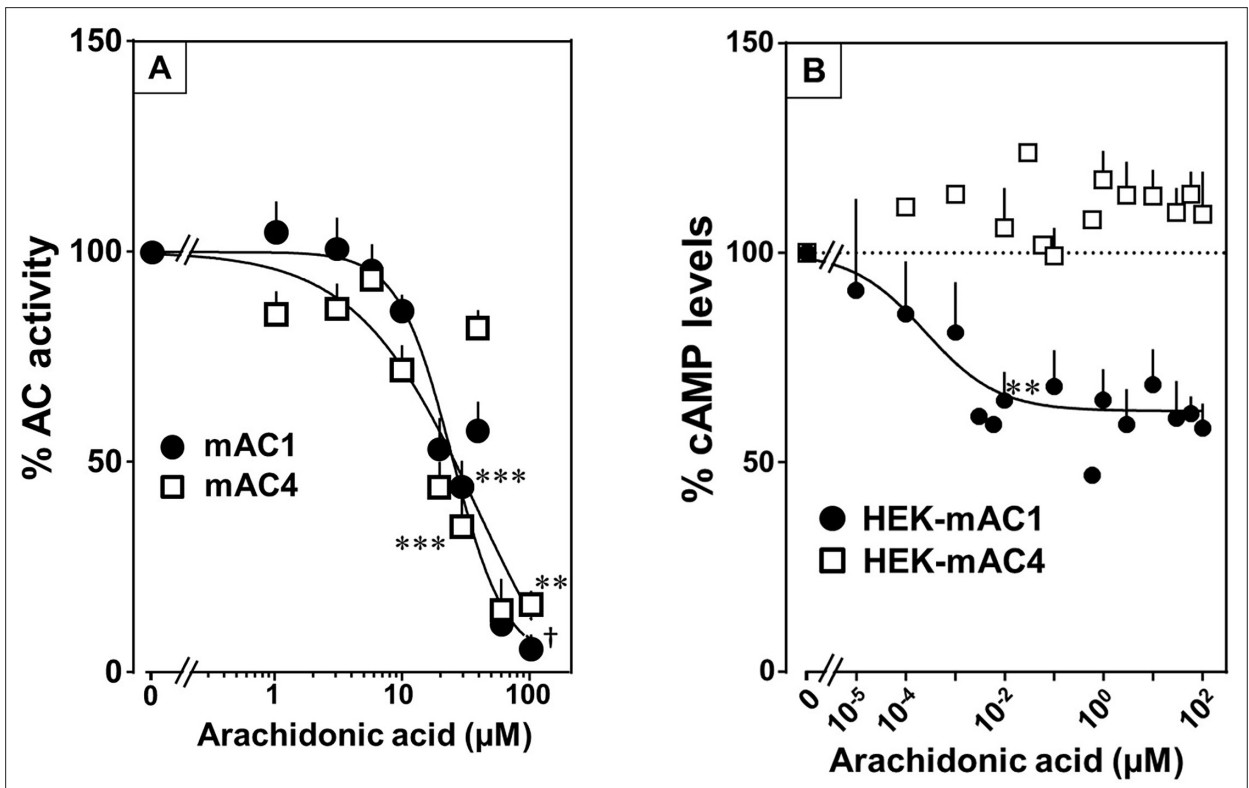

**Figure 6.** Arachidonic acid attenuates 300 nM Gsα-stimulated activities of mACs 1 and 4. (**A**) Arachidonic acid attenuates Gsα-stimulated mACs 1 and 4. Basal and Gsα-stimulated activities of mAC1 were 0.12 ± 0.01 and 0.42 ± 0.03 and of mAC4 0.02 ± 0.002 and 0.14 ± 0.02 nmol cAMP•mg$^{-1}$•min$^{-1}$, respectively. IC$_{50}$ of arachidonic acid for mAC1 and mAC4 were 23 and 36 μM, respectively. $n$ = 3–9. (**B**) Effect of arachidonic acid on HEK-mAC1 and HEK-mAC4 cells. Cells were stimulated by 10 μM isoproterenol (set as 100 %) in the presence of 0.5 mM IBMX (3-isobutyl-1-methylxanthine). Basal and isoproterenol-stimulated cAMP levels in HEK-mAC1 were 1.03 ± 0.15 and 1.66 ± 0.28 and in HEK-mAC4 0.20 ± 0.04 and 0.86 ± 0.24 pmol cAMP/10,000 cells, respectively. IC$_{50}$ for HEK-mAC1 was 250 pM. $n$ = 2–11, each with three replicates. Data are mean ± SEM. One-sample $t$ tests were performed. Significances: **$p < 0.01$; ***$p < 0.001$; †$p < 0.0001$. For clarity, not all significances are indicated.

The online version of this article includes the following source data and figure supplement(s) for figure 6:

**Source data 1.** Including data used for generating *Figure 6A, B*.

**Figure supplement 1.** Effect of 20 μM lipids on 300 nM Gsα-stimulated mAC1.

**Figure supplement 1—source data 1.** Including data used for generating *Figure 6—figure supplement 1*.

**Figure supplement 2.** Effect of 20 μM lipids on 300 nM Gsα-stimulated mAC4.

**Figure supplement 2—source data 1.** Including data used for generating *Figure 6—figure supplement 2*.

**Figure supplement 3.** Palmitoleic acid inhibits mACs 1 and 4 stimulated by 300 nM Gsα.

**Figure supplement 3—source data 1.** Including data used for generating *Figure 6—figure supplement 3*.

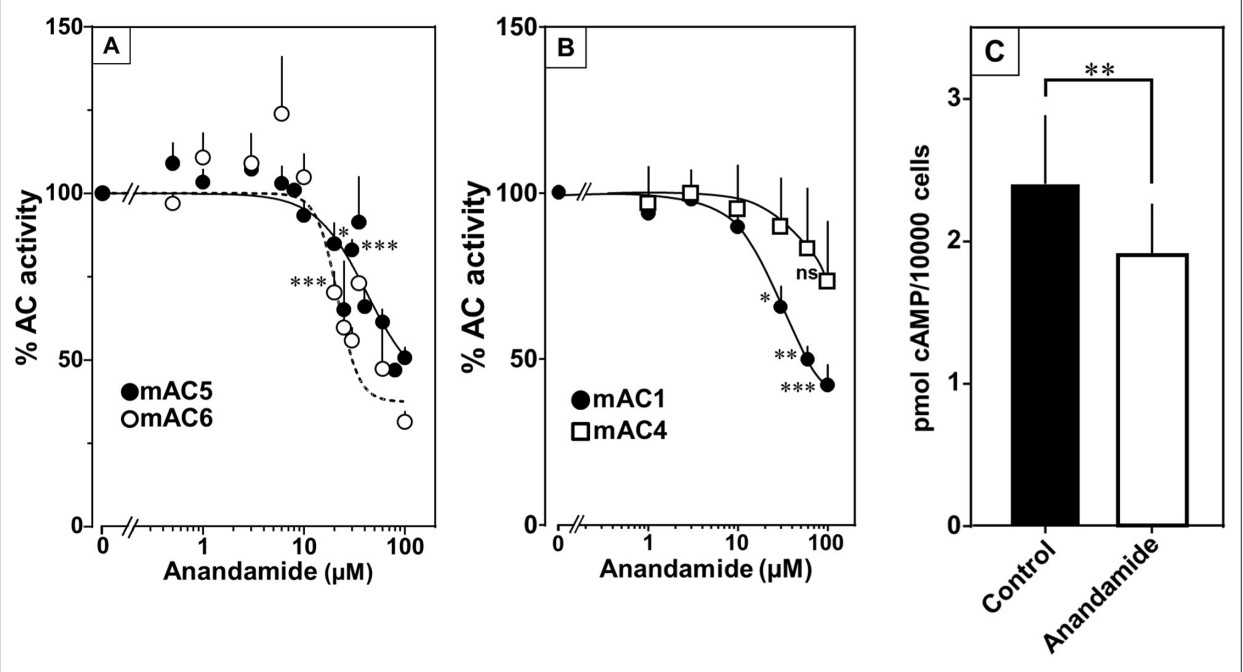

**Figure 7.** Anandamide attenuates 300 nM Gsα-stimulated activities of mACs 1, 4, 5, and 6. (**A**) Effect of anandamide on Gsα-stimulated mAC5 and 6. Basal and Gsα activities of mAC5 were 0.05 ± 0.01 and 0.98 ± 0.12 and of mAC6 0.05 ± 0.01 and 0.78 ± 0.12 nmol cAMP•mg$^{-1}$•min$^{-1}$, respectively. IC$_{50}$ of anandamide were 42 and 22 μM, respectively. $n$ = 3–32. (**B**) Anandamide attenuates mAC1 but not mAC4 stimulated by Gsα. Basal and Gsα-stimulated activities of mAC1 were 0.12 ± 0.01 and 0.40 ± 0.03 and of mAC4 0.02 ± 0.002 and 0.15 ± 0.02 nmol cAMP•mg$^{-1}$•min$^{-1}$, respectively. IC$_{50}$ for mAC1 was 29 μM. $n$ = 3–4, each with two technical replicates. (**C**) Effect of anandamide on 2.5 μM isoproterenol-stimulated HEK-mAC5. Basal and isoproterenol-stimulated cAMP levels of HEK-mAC5 were 1.8 ± 0.22 and 2.4 ± 0.48 pmol cAMP/10,000 cells, respectively. The control bar represents 2.5 μM isoproterenol stimulation alone. $n$ = 5–6, each with three technical replicates. IC$_{50}$ of anandamide was 133 μM. Data are mean ± SEM. One-sample $t$ tests (A, B) and paired $t$ test (C) were performed. Significances: ns: not significant p > 0.05; *p < 0.05; **p < 0.01; ***p < 0.001. For clarity, not all significances are indicated.

The online version of this article includes the following source data and figure supplement(s) for figure 7:

**Source data 1.** Including data used for generating *Figure 7A–C*.

**Figure supplement 1.** Effect of 20 μM lipids on 300 nM Gsα-stimulated mAC5.

**Figure supplement 1—source data 1.** Including data used for generating *Figure 7—figure supplement 1*.

**Figure supplement 2.** Effect of 20 μM lipids on 300 nM Gsα-stimulated mAC6.

**Figure supplement 2—source data 1.** Including data used for generating *Figure 7—figure supplement 2*.

substituted by those of mAC5 (design in *Figure 9A*, right). The intention was to obtain a chimera, mAC5$_{(membr)}$–AC3$_{(cat)}$, with a loss of receptor function, i.e., no enhancement by oleic acid, and a gain of another receptor function, i.e., attenuation of activity by anandamide. Successful expression and membrane insertion of the chimera in HEK293 cells was demonstrated by specific conjugation to Cy5.5 fluorophore, using the protein ligase Connectase (*Figure 9A*, left; *Fuchs, 2023*). cAMP synthesis of isolated membranes from these cells was stimulated up to 10-fold by addition of 300 nM Gsα, comparable to membranes with recombinant mAC3 or mAC5 proteins (*Figure 9B*). mAC activity in the mAC5$_{(membr)}$–AC3$_{(cat)}$ chimera was not enhanced by oleic acid, i.e., loss of receptor function, but was attenuated by anandamide, i.e., gain of receptor function (*Figure 9C, D*). The attenuation was comparable to results obtained with mAC5 membranes, i.e., IC$_{50}$ was 29 μM mAC5$_{(membr)}$–AC3$_{(cat)}$ (*Figure 9E*) compared to 42 μM for mAC5. This means that the attenuating receptor property of mAC5 was successfully grafted onto the mAC3-catalytic dimer. We take this to support the hypothesis that the mammalian mAC membrane domains operate as receptors using lipid ligands. The data virtually rule out unspecific lipid effects such as disturbance of membrane integrity by intercalation and surfactant or detergent effects. In addition, the data demonstrated that the signal most likely originates from the receptor entity and is transmitted through the subsequent linker regions to the catalytic dimer.

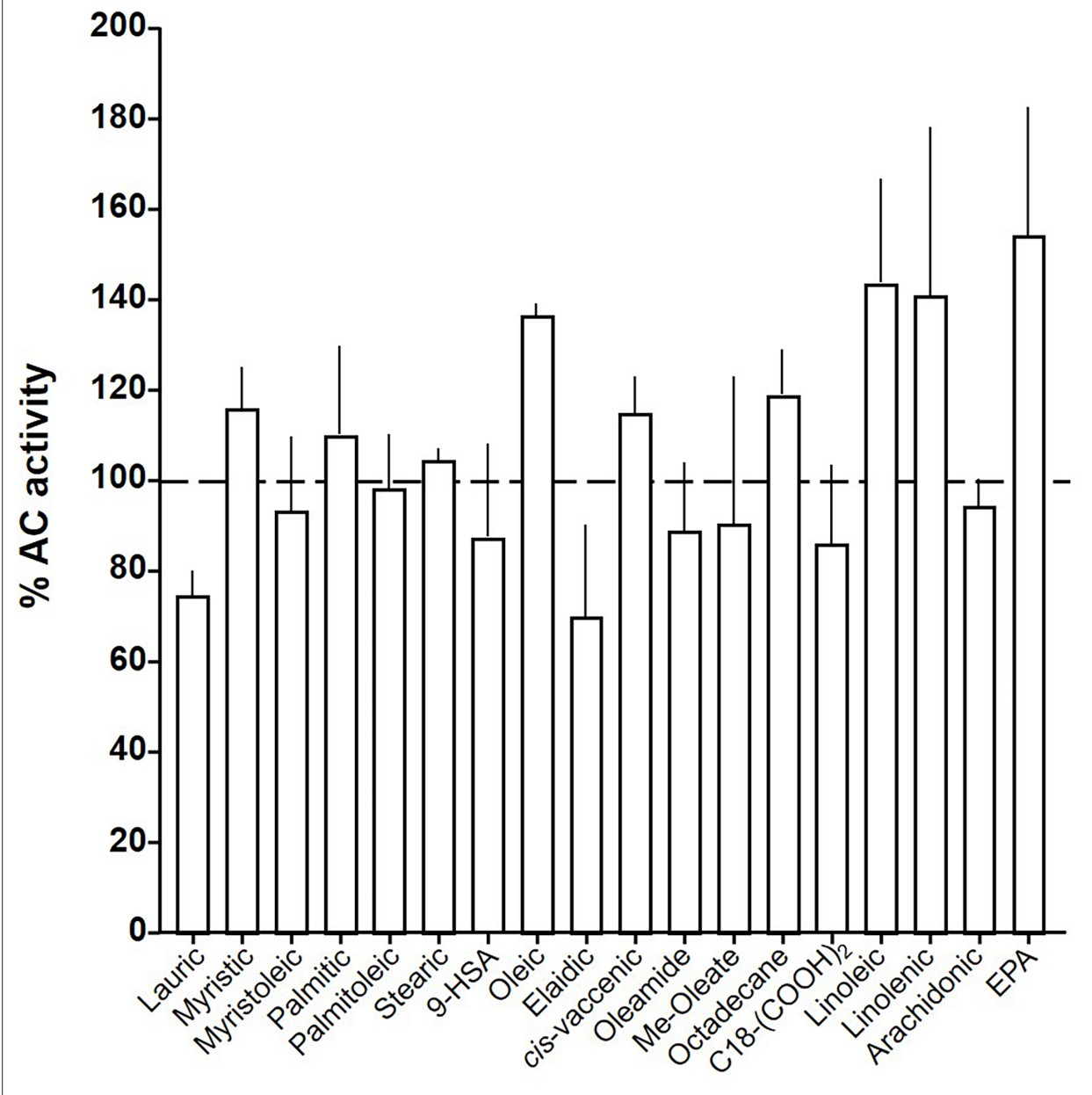

**Figure 8.** Effect of 20 µM lipids on 300 nM Gsα-stimulated mAC8. Basal and Gsα activities were 0.19 ± 0.01 and 1.04 ± 0.19 nmol cAMP•mg$^{-1}$•min$^{-1}$, respectively. Error bars denote SEM of $n$ = 2–5.

The online version of this article includes the following source data for figure 8:

**Source data 1.** Including data used for generating *Figure 8*.

The findings were further substantiated in vivo using HEK293-mAC5$_{(membr)}$–AC3$_{(cat)}$ cells. cAMP formation primed by 2.5 µM isoproterenol was attenuated by anandamide in HEK293-mAC5$_{(membr)}$–AC3$_{(cat)}$ cells by 66%, (*Figure 10*), i.e., a gain of function which remarkably exceeded the anandamide attenuation in HEK293-mAC5 cells of 23%. In HEK293-mAC5$_{(membr)}$–AC3$_{(cat)}$ cells oleic acid was ineffective, i.e., loss of function (data not shown). The results bolster the notion that mAC isoforms are receptors with lipids as ligands.

Lastly, we prepared membranes from mouse brain cortex in which predominantly mAC isoforms 2, 3, and 9 are expressed, isoforms with demonstrated enhancement of Gsα stimulation by oleic acid (*Sanabra and Mengod, 2011*). In cortical membranes 20 µM oleic acid enhanced Gsα-stimulated cAMP formation 1.5-fold with an EC$_{50}$ of 5 µM, almost identical to the one determined for mAC2, 3,

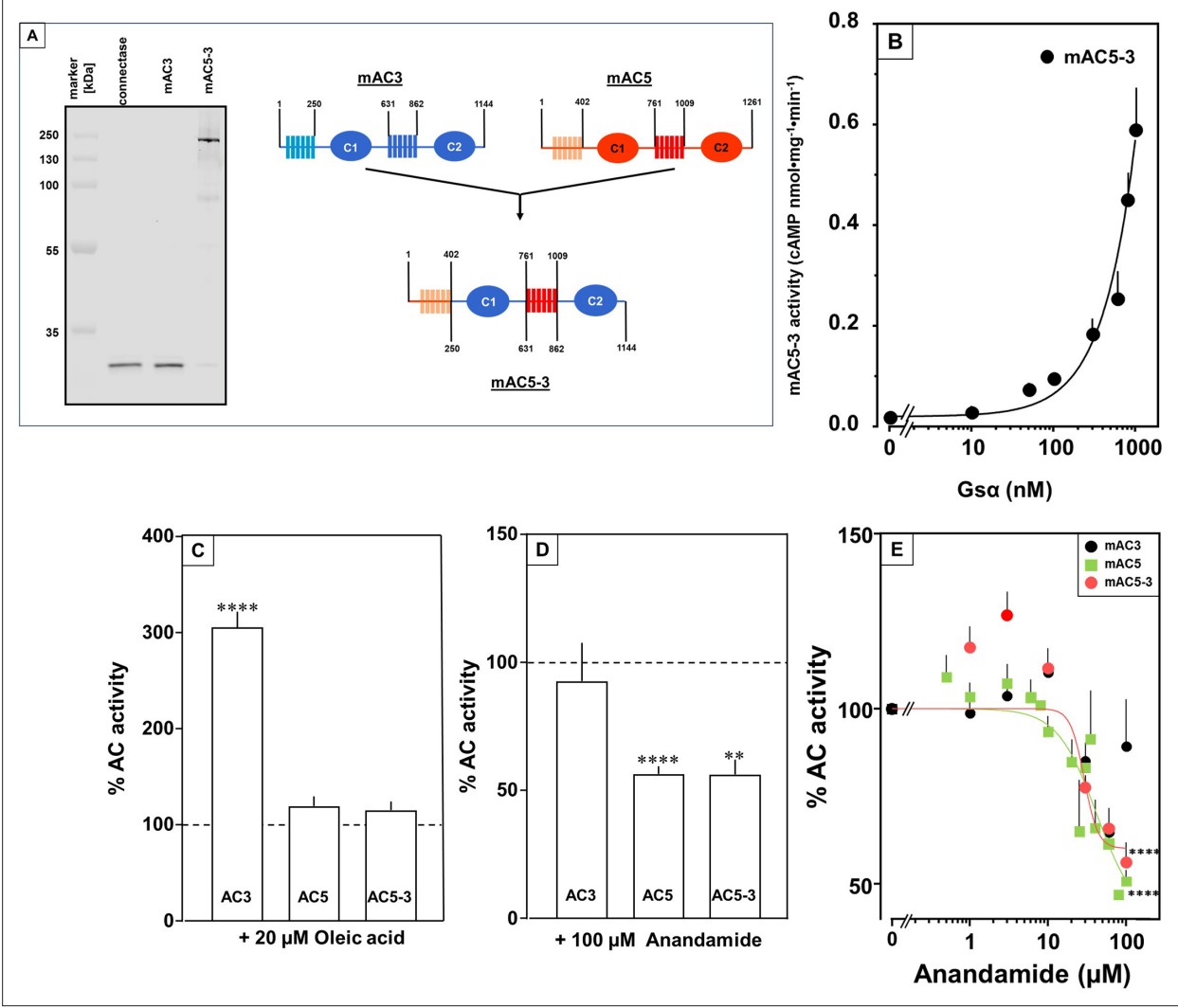

**Figure 9.** Receptor properties are exchangeable between mAC isoforms. (**A**, left) Detection of AC5(membr)–AC3(cat) receptor chimeras. AC5(membr)–AC3(cat) (AC5–3) (**Tews et al., 2005**) was expressed in HEK293 cells with an N-terminal tag for labeling with the protein ligase Connectase. The membrane preparation was incubated with fluorophore-conjugated Connectase and separated by sodium dodecyl sulfate–polyacrylamide gel electrophoresis (SDS–PAGE). A fluorescence scan of the gel detects AC5(membr)–AC3(cat) (right), the reagent (fluorophore-conjugated Connectase) is detected when using HEK293 membrane (middle) or a buffer control (left); (**A**, right) Design of the chimeric AC5–3 construct. Numbers are amino acid positions in mAC3 and 5, respectively. (**B**) Gsα concentration–response curve of mAC5–3. Basal activity for mAC5–3 was 0.02 pmol cAMP•mg$^{-1}$•min$^{-1}$. Error bars denote SEM of $n = 3$, each with two technical replicates. (**C**) Effect of 20 µM oleic acid on 300 nM Gsα-stimulated mACs 3, 5, and 5–3. Basal and Gsα activities of mACs 3, 5, and 5–3 were 0.02 ± 0.003 and 0.11 ± 0.02, 0.05 ± 0.01 and 0.98 ± 0.12, and 0.01 ± 0.004 and 0.2 ± 0.02 nmol cAMP•mg$^{-1}$•min$^{-1}$, respectively. $n = 7–33$. (**D**) Effect of 100 µM Anandamide on 300 nM Gsα-stimulated mACs 3, 5, and 5–3. Basal and Gsα activities of mACs 3, 5, and 5–3 were 0.02 ± 0.002 and 0.19 ± 0.02, 0.05 ± 0.01 and 0.98 ± 0.12, and 0.02 ± 0.003 and 0.23 ± 0.04 nmol cAMP•mg$^{-1}$•min$^{-1}$, respectively. $n = 6–9$. IC$_{50}$ for mAC5 and mAC5–3 were 42 and 29 µM, respectively. (**E**). Exchange of TM domains transfers anandamide effect on mAC3. Basal and Gsα-stimulated activities of mAC3 were 0.02 ± 0.002 and 0.12 ± 0.02 nmol cAMP•mg$^{-1}$•min$^{-1}$, respectively. Basal and Gsα-stimulated activities of mAC5 were 0.05 ± 0.005 and 0.98 ± 0.12 nmol cAMP•mg$^{-1}$•min$^{-1}$, respectively. Basal and Gsα-stimulated activities of mAC5–3 were 0.02 ± 0.002 and 0.22 ± 0.03 nmol cAMP•mg$^{-1}$•min$^{-1}$, respectively. Calculated IC$_{50}$ concentrations of anandamide for mAC5 and mAC5–3 were 42 and 29 µM, respectively. Data are mean ± SEM. One-sample $t$ tests. Significances: **$p < 0.01$; ****$p < 0.0001$.

The online version of this article includes the following source data for figure 9:

**Source data 1.** PDF file containing original gel for **Figure 9A**, indicating the relevant bands.

**Source data 2.** Original file containing gel for **Figure 9A**.

**Source data 3.** Including data used for generating **Figure 9B–E**.

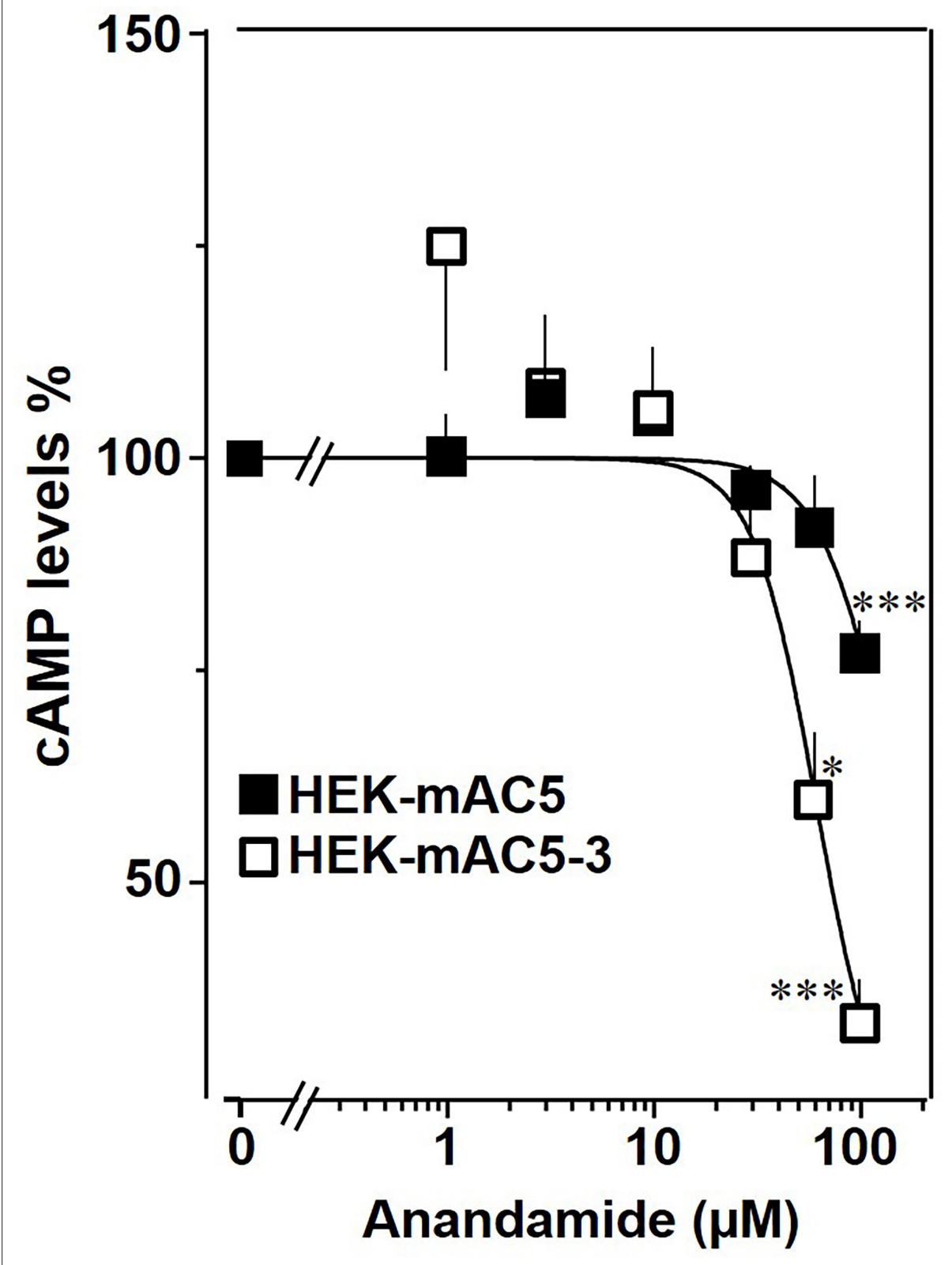

**Figure 10.** mAC5–mAC3 receptor transfer analyzed in HEK293 cells. Effect of anandamide on HEK-mAC5 and HEK-mAC5–3 cells stimulated by 2.5 µM isoproterenol (set as 100%). Basal and isoproterenol-stimulated cAMP levels in HEK-mAC5 were 1.80 ± 0.22 and 2.29 ± 0.39 and in HEK-mAC5–3 (+0.5 mM IBMX) 0.17 ± 0.02 and 3.11 ± 0.55 pmol cAMP/10,000 cells, respectively. $n$ = 4–11. IC$_{50}$ for HEK-mAC5 and HEK-mAC5–3 were 133 and 60 µM,

*Figure 10 continued on next page*

*Figure 10 continued*

respectively. Anandamide had no effect on basal activity of HEK-mAC5 and stimulated HEK-mAC3 cells in concentrations up to 100 µM (data not shown). Data are mean ± SEM. One-sample *t* tests were performed. Significances: *p < 0.05; ***p < 0.001.

The online version of this article includes the following source data for figure 10:

**Source data 1.** Including data used for generating *Figure 10*.

7, and 9 (*Figure 11*). This suggests that mACs in brain cortical membranes are similarly affected by fatty acids.

## Discussion

In the past, the biology of the two membrane anchors of mACs, highly conserved in an isoform-specific manner, remained unresolved. The theoretical possibility of a receptor function of these large hexahelical anchor domains, each comprising 150–170 amino acids, was considered unlikely (*Schuster et al., 2024*). Our data are a transformative step toward resolving this issue and introduce lipids as critical participants in regulating cAMP biosynthesis in mammals. The first salient discovery is the identification of the membrane domains of mACs as a new class of receptors for chemically defined ligands which set the level of stimulation by the GPCR/Gsα system. This conclusion is based on (1) the dodecahelical membrane domains of the nine mAC receptors have distinct, conserved isoform-specific sequences for the TM1 and TM2 domains *Schultz, 2022*; (2) the receptors have distinct ligand specificities and affinities in the lower micromolar range; (3) isoform dependently ligands either enhance or attenuate Gsα-stimulated mAC activities; (4) receptor properties are transferable between isoforms by interchanging membrane domains; (5) isoproterenol-stimulated formation of cAMP in vivo is affected by addition of extracellular ligands; (6) Gsα-stimulated cAMP formation in mouse cortical membranes is enhanced by oleic acid. Therefore, the results establish a new class of receptors, the membrane domains of mACs, with lipids as ligands. The data question the utility of the currently used mAC subclassification, which groups mAC1, 3, 8, mAC2, 4, 7, mACs 5, 6, and mAC9 together (*Dessauer et al., 2017*). At this point, mAC 1, 4, 5, and 6, which are ligand-attenuated, may be grouped together and a second group may consist of isoforms 2, 3, 7, and 9 which are ligand-enhanced. Our data do not contradict earlier findings concerning regulation of mACs via GPCRs, cellular localization of mAC isoforms or regional cAMP signaling (*Dessauer et al., 2017*). Instead, the data reveal a completely new level of regulation of cAMP biosynthesis in which two independent modalities of signaling, i.e., direct, tonic lipid signaling and indirect phasic signaling via the GPCR/Gsα circuits intersect at the crucial biosynthetic step mediated by the nine mAC isoforms.

The second important finding is the observation that the extent of enhancement of mAC3 activity by 20 µM oleic acid is uniform up to 1000 nM Gsα (*Figure 2*, right). We suppose that in mAC3 the equilibrium of two differing ground states favors a Gsα unresponsive state and the effector oleic acid concentration dependently shifts this equilibrium to a Gsα responsive state (*Seth et al., 2020*). In contrast, the equilibrium of ground states of mAC5 probably is opposite, i.e., the one accessible to Gsα stimulation predominates and stimulation by Gsα is maximal. In addition, oleic acid has little effect because the mAC5 receptor domain does not bind oleic acid (*Figure 1D*, right, and *Figure 2*, center). A ligand for mAC5, e.g., anandamide or arachidonic acid, likely shifts the equilibrium of ground states to a Gsα unresponsive state and inhibits stimulation. The biological balance of ground states appears to be an intrinsic property which is isoform specifically imprinted in mACs. Probably, it defines a major element of regulation and enables distinct inhibitory or stimulatory inputs by extracellular lipid ligands. The ground states probably are separated by a low transition energy and are stabilized by receptor occupancy. Hitherto available structures required Gsα and/or forskolin for stabilization and probably did not capture different ground states (*Schuster et al., 2024*; *Qi et al., 2019*; *Qi et al., 2022*; *Sunahara et al., 1997b*; *Tesmer and Sprang, 1998*; *Vercellino et al., 2017*). Mechanistically, tonic levels of lipid ligands affect the ground states and thus set the bounds of cAMP formation elicited by phasic GPCR/Gsα stimulation. As such lipid signaling through the mAC membrane receptors appears to represent a higher level of a systemic regulatory input based on constant monitoring the physiological and nutritional status of an organism.

Lipid signaling is much less characterized than solute signaling (*Eyster, 2007*). Most of the highly functionalized ligands for GPCRs are storable in vesicles and the release, inactivation and removal

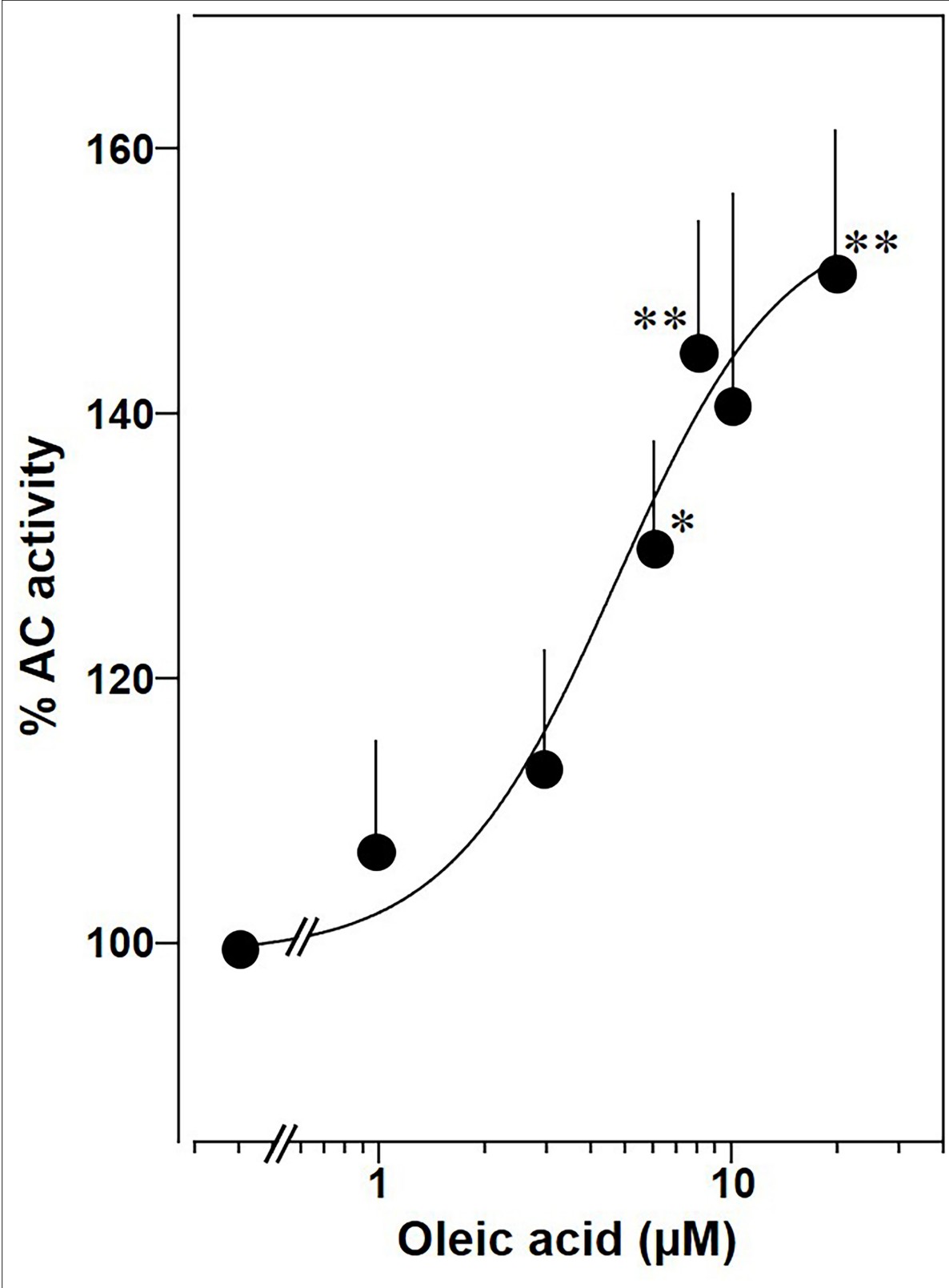

**Figure 11.** Oleic acid concentration dependently potentiates mAC activity in brain cortical membranes from mouse. Basal and 300 nM Gsα activities were 0.4 ± 0.1 and 2.7 ± 0.7 nmol cAMP•mg$^{-1}$•min$^{-1}$, respectively. $n$ = 4–6. One-sample $t$ test: *p < 0.05; **p < 0.01 compared to 100% (300 nM Gsα stimulation).

*Figure 11 continued on next page*

*Figure 11 continued*

The online version of this article includes the following source data for figure 11:

**Source data 1.** Including data used for generating *Figure 11*.

are strictly controlled. On the other hand, the very nature of lipids, i.e., high flexibility of aliphatic chains, low water solubility, propensity for nonspecific protein binding, membrane permeability and potential effects on membrane fluidity complicate discrimination between extra- and intracellular lipid actions (*Samovski et al., 2023*). Yet, viewed from an evolutionary perspective, lipids possibly are ideal primordial signaling molecules because for the emergence of the first cell lipids were required to separate an intra- and extracellular space. Conceivably, lipids derived from membrane lipids were used for regulatory purposes early-on. In association with the evolution of bacterial mAC progenitors lipid ligands may have persisted in evolution and regulation by GPCR/Gsα in metazoans was acquired and expanded later.

The concentrations of free fatty acids in serum or interstitial fluid usually are rather low, mostly below the $EC_{50}$ concentrations determined in this study (*Grundmann et al., 2021*; *Ulven and Christiansen, 2015*; *Huber and Kleinfeld, 2017*). This raises the question of the origin of lipid ligands under physiological conditions. It is well known that cell membranes are highly dynamic (*Alam et al., 1995*; *McMahon and Gallop, 2005*). Within limits, cell morphology and lipid composition are in constant flux to accommodate diverse functional requirements. Remodeling of cell membranes is accomplished by targeted phospholipid biosynthesis and by regulated lipolysis of membrane lipids, e.g., by phospholipase $A_2$, mono- and diacylglycerol lipases or lipoprotein lipases (*Brown et al., 2003*). Therefore, a speculative possibility is that lipid ligands are acutely and locally generated directly from membrane lipids (*Liu et al., 2006*; *Muccioli, 2010*). Such regulation probably happens at the level of individual cells, cellular networks, complex tissues, and whole organs. Additional potential lipid sources available for ligand generation may be, among others, lipids in nutrients (*Alam et al., 1995*), exosomes present in blood, serum lipids, chylomicrons, blood triglycerides, and lipids originating from the microbiome. On the one hand, the discovery of lipids as ligands for the mAC receptors broadens the basis of regulation of cAMP biosynthesis with potentially wide-ranging consequences in health and disease. On the other hand, the data pose the challenge to identify how the tonic signal is generated and regulated.

## Materials and methods
### Reagents and materials
ATP, creatine kinase, and creatine phosphate were from Merck. Except for lauric acid (Henkel) and 1,18-octadecanedicarboxylic acid (Thermo Fisher Scientific), lipids were from Merck. 10 mM stock solutions were prepared in analytical grade Dimethyl sulfoxide (DMSO) and kept under nitrogen. For assays the stock solution was appropriately diluted in 20 mM 3-(N-morpholino)propanesulfonic acid (MOPS) buffer, pH 7.5, suitable to be added to the assays resulting in the desired final concentrations. The final DMSO concentrations in in vitro and in vivo assays were maximally 1%, a concentration without any biochemical effect as checked in respective control incubations. The constitutively active GsαQ227L mutant protein was expressed and purified as described earlier (*Graziano et al., 1989*; *Graziano et al., 1991*; *Sunahara et al., 1997a*).

### General experimental procedures
For HPLC analysis, a Waters HPLC system (1525 pump, 2996 photodiode array detector, 7725i injector, 200 series PerkinElmer vacuum degasser) was used. Solvents were HPLC or LC–MS grade from Merck-Sigma. One-dimensional $^1$H- and $^{13}$C-NMR spectra were recorded on a 400 MHz Bruker AVANCE III NMR spectrometer equipped with a 5-mm broadband SmartProbe and AVANCE III HD Nanobay console. Spectra were recorded in methanol-$d_4$ and calibrated to the residual solvent signal ($\delta_H$ 3.31 and $\delta_C$ 49.15 ppm).

### Lung tissue extraction and fractionation
1.24 kg bovine lung was minced in a meat grinder, then mixed and homogenized with 1.2 l 50 mM MOPS, pH 7.5, in a Waring blender (4°C) resulting in 2.3 l homogenate. It was centrifuged (30 min,

4°C, 7200 × $g$) resulting in 1.2 l supernatant. The pH was adjusted to 1 using 7% HCl. Equal volumes of $CH_2Cl_2$/MeOH (2:1) were mixed with the supernatant in a separatory funnel and shaken vigorously. Centrifugation was at 5300 × $g$ for 30 min. The lower organic $CH_2Cl_2$ layer was recovered and the solvent was evaporated yielding 2 g of dried material. It was dissolved in 100 ml petroleum ether and subjected to normal-phase silica gel vacuum liquid chromatography (60 H Supelco). The column was eluted stepwise with solvents of increasing polarity from 90:10 petroleum ether/EtOAc to 100% EtOAc, followed by 100% MeOH. 17 fractions (A–Q) of 300 ml were collected and dried. Fraction E (eluting at 40:60 petroleum ether/EtOAc) was analyzed by RP-HPLC using a linear MeOH/$H_2O$ gradient from 80:20 to 100:0 (0.1% TFA Trifluoroacetic acid) for 15 min, followed by 100:0 for 30 min (Knauer Eurosphere II C18P 100-5, 250 × 8 mm, 1.2 ml/min flow rate, UV absorbance monitored at 210 nm) to yield five subfractions: E1–E5. Fraction E2 was analyzed by $^1$H- and $^{13}$C-NMR which indicated the presence of aliphatic lipids and fatty acids (*Figure 1—figure supplement 3*).

## Gas chromatography–mass spectrometry analysis

Fraction E2 was analyzed by gas chromatography–mass spectrometry. Acids were acid trimethylsilylated using *N,O*-bis(trimethylsilyl)trifluoroacetamide + trimethylchlorosilane (99:1 vol/vol). The mixture was heated for 2 hr at 90°C. After cooling and clearing the sample was transferred into a GC vial in 200 µl hexane.

An Agilent Technologies GC system (8890 gas chromatograph and 5977B mass spectrometer equipped with a DB-HP5MS UI column, 30 m × 0.25 mm, film thickness of 0.25 µm) was used. Injection volume was 1 µl. The temperature was kept at 100°C for 5 min, and then increased at 53°C/min to 240°C. The rate was decreased to 3°C/min to reach 305°C. Carrier gas was $He_2$ (99.9%; 1.2 ml/min). Ionization was with 70 eV and MS spectra were recorded for a mass range $m/z$ 35–800 for 35 min. Compounds were identified by comparing the spectra with those in the NIST library. Individual compound content is given as a relative % of the total peak area.

## Plasmid construction and protein expression

hAC sequences were from NCBI were: ADCY1: NM_021116.3; ADCY2: NM_020546.2; ADCY3: NM_004036.4; ADCY4: NM_001198568.2; ADCY5: NM_183357.2; ADCY6: NM_015270.4; ADCY7: NM_001114.4; ADCY8: NM_001115.2; ADCY9: NM_001116.3. Human mAC genes were from GenScript and fitted with a C-terminal FLAG-tag. The chimera mAC5(TM)_mAC3(cat) had an N-terminal connectase-tag, MPGAFDADPLVVEIAAAGA, followed by AC5(1–402)_AC3(250–631)_AC5(761–1009)_AC3(862–1144). The gene was synthesized by GenScript. HEK293 cells were maintained in Dulbeccos Modified Eagle Medium (DMEM) with 10% fetal bovine serum at 37°C and 5% $CO_2$. Transfection with AC plasmids was with PolyJet (SignaGen, Frederick, MD, USA). Permanent cell lines were generated by selection for 7 days with 600 µg/ml G418 and maintained with 300 µg/ml For membrane preparation, cells were tyrpsinized, collected by centrifugation (3000 × $g$, 5 min) and lysed and homogenized in 20 mM HEPES, pH 7.5, 1 mM Ethylenediaminetetraacetic acid (EDTA), 2 mM $MgCl_2$, 1 mM Dithiothreitol (DTT), one tablet of cOmplete, EDTA-free (per 50 ml) and 250 mM sucrose by 20 strokes in a potter homogenizer on ice. Debris was removed (5 min at 1000 × $g$, 0°C), membranes were collected at 100,000 × $g$, 60 min at 0°C, suspended and stored at −80°C in 20 mM MOPS, pH 7.5, 0.5 mM EDTA, 2 mM $MgCl_2$. Membrane preparation from mouse brain cortex was according to *Seth et al., 2020*; *Schultz and Schmidt, 1987*. Three cerebral cortices were dissected and homogenized in 4.5 ml cold 48 mM Tris–HCl, pH 7.4, 12 mM $MgC1_2$, and 0.1 mM Ethylene glycol-bis(β-aminoethyl ether)-N,N,N′,N′-tetraacetic acid (EGTA) with a Polytron hand disperser (Kinematica AG, Switzerland). The homogenate was centrifuged for 15 min at 12,000 × $g$ at 4°C. The pellet was washed once with 5 ml 1 mM $KHCO_3$. The final suspension in 2 ml 1 mM $KHCO_3$ was stored in aliquots at −80°C.

## DNA extraction

DNA from 1 × $10^6$ cells of permanently transfected and non-transfected HEK293 cells was extracted using the High Pure PCR Template Preparation Kit (Roche) according to the manufacturer's instructions. DNA concentrations were determined at 260 nm using a sub-microliter cell (IMPLEN) in a P330 NanoPhotometer (IMPLEN). Elution buffer (Roche) was used for blanks.

## Polymerase chain reaction

100 ng of template DNA was mixed with 0.5 µM Forward primer and 0.5 µM Reverse primer. 12.5 µl 2X KAPA2G Fast (HotStart) Genotyping Mix with dye and water was added, total reaction volume 25 µl according to the KAPA2G Fast HotStart Genotyping Mix kit (Roche) protocol. PCR followed the cycling protocol in a Biometra T3000 thermocycler:

| Step | Temperature (°C) | Duration | Cycles |
| --- | --- | --- | --- |
| Initial denaturation | 95 | 3 min | 1 |
| Denaturation | 95 | 15 s | 35 |
| Annealing | 60 | 15 s | |
| Extension | 72 | 30–60 s | |
| Final extension | 72 | 2–4 min | 1 |

Extension and final extension times were adjusted to the expected amplicon length.

The PCR products were directly loaded on a 1.5% agarose gel. A 1 kb DNA ladder (New England Bio Labs #N3232S) was mixed with Gel Loading Dye Purple 6X (New England Biolabs #B7024S) and water. After running the gel for 15–20 min at 90 V in 1× Tris(hydroxymethyl)aminomethane-acetate-ethylenediaminetetraacetic acid (TAE) buffer, the gel was stained in an Ethidium bromide bath and left running for another 10–20 min. The gels were then evaluated under UV light in a UVP GelStudio PLUS (Analytik Jena) gel imager.

| AC isoform | Forward primer (5'–3') | Reverse primer (5'–3') |
| --- | --- | --- |
| 1 | GTCAACAGGTACATCAGCCGCC | AGCCTCCTTCCCAGCTGCTGC |
| 2 | AGGAGACTGCTACTACTGTGTATCTGGAC | GGATGCCACGTTGCTCTGGGA |
| 3 | TTCATCCTGGTGATGGCAAATGTCGT | GGAGTTGTCCACCACCTGGTG |
| 4 | CGGGGATGCCAAGTTCTTCCAGGTCATTG | GCCTAGGGTAGCTGAAGGAGG |
| 5 | CCTCATCCTGCGCTGCACCCAGAAGCG | ACTGAGC |
| 6 | TCCTGAGCCGTGCCATCGA | ACTGCTGGGGCCCCCATTGAG |
| 7 | TCCTCGGCGACTGCTACTACTG | GTTCAGCCCCAGCCCCTGAAA |
| 8 | ACTTGCGGAGTGGCGATAAATTGAGA | TGGCAAATCAGATTTGTCGGTGCC |
| 9 | CGCTGTGCTTCCTCCTGGTG | CACACTCTTTGAAACGTTGAGC |

## AC assay

In a volume of 10 µl, AC activities were measured using 1 mM ATP, 2 mM MgCl$_2$, 3 mM creatine phosphate, 60 µg/ml creatine kinase, and 50 mM MOPS pH 7.5. The cAMP assay kit from Cisbio (Codolet, France) was used for detection according to the supplier's instructions. For each assay, a cAMP standard curve was established. EC$_{50}$ and IC$_{50}$ values were calculated by GraphPad Prism version 8.4.3 for Windows, GraphPad Software, San Diego, CA, USA, https://www.graphpad.com .

## cAMP accumulation assay

HEK293 cells stably expressing mAC isoforms 3, 5, and mAC5(membr)_mAC3(cat) were plated at 2500–10,000 cells/well into 384-well plates. Cells were treated with varying concentrations of lipids and incubated for 10 min at 37°C and 5% CO$_2$. 2.5–10 µM isoproterenol was added to stimulate cAMP formation and cells were incubated for 5 min. HEK293-AC5–3 was assayed in the presence of the phosphodiesterase inhibitor 0.5 mM isobutyl-methyl-xanthine. Addition of Cisbio HTRF detection reagents stopped the reaction and cAMP levels were determined.

## Data handling and analysis

Experimental results were evaluated using GraphPad Prism version 8.4.3. Assays were conducted with a minimum of two technical replicates from at least two independent assays, as specified in the

figure legends. Average values from duplicate or triplicate experiments were designated as single points and data are expressed as means ± SEM. Graphs were generated by GraphPad and assembled in PowerPoint. Outliers were identified using the 'Identify Outliers' function of GraphPad (ROUT method).

## Acknowledgements

We are indebted to Prof. Andrei Lupas, Max-Planck-Institute of Biology, Tübingen, for support, advice, and critique. We thank to N Grzegorzek, Organic Chemistry, for GC/MS measurements. Funding was from the Deutsche Forschungsgemeinschaft (Schu275/45-1) and from institutional funds from the Max-Planck-Gesellschaft.

## Additional information

### Funding

| Funder | Grant reference number | Author |
| --- | --- | --- |
| Deutsche Forschungsgemeinschaft | Schu275/45-1 | Joachim E Schultz |
| Max-Planck-Gesellschaft | institutional funds | Adrian CD Fuchs |

The funders had no role in study design, data collection, and interpretation, or the decision to submit the work for publication.

### Author contributions

Marius Landau, Data curation, Investigation, Methodology, Validation, Visualization, Writing – review and editing; Sherif Elsabbagh, Data curation, Investigation, Validation, Visualization, Writing – review and editing; Harald Gross, Data curation, Investigation, Methodology, Writing – review and editing; Adrian CD Fuchs, Data curation, Visualization; Anita CF Schultz, Formal analysis, Data curation, Visualization; Joachim E Schultz, Conceptualization, Formal analysis, Supervision, Writing - original draft, Project administration, Writing – review and editing

### Author ORCIDs

Marius Landau ⓘ https://orcid.org/0000-0002-4638-1908
Adrian CD Fuchs ⓘ https://orcid.org/0000-0001-6550-1795
Joachim E Schultz ⓘ https://orcid.org/0000-0002-1985-4853

Reviewer #1 (Public review): https://doi.org/10.7554/eLife.101483.3.sa1
Reviewer #3 (Public review): https://doi.org/10.7554/eLife.101483.3.sa2
Author response https://doi.org/10.7554/eLife.101483.3.sa3

## Additional files

### Supplementary files

• MDAR checklist

### Data availability

All data generated or analyzed during this study are included in the manuscript and supporting files.

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
