## [Editor Report · eLife Assessment]

This manuscript describes an **important** study of a new lipid-mediated regulation mechanism of adenylyl cyclases. The biochemical evidence provided is **convincing** and will trigger more research in this mechanism. This manuscript will be of interest to all scientists working on lipid regulation and adenylyl cyclases.

---

## [Referee Report · Reviewer #1 (Public review)]

Summary:

The authors show that the Gαs-stimulated activity of human membrane adenylyl cyclases (mAC) can be enhanced or inhibited by certain unsaturated fatty acids (FA) in an isoform-specific fashion. Thus, with IC50s in the 10-20 micromolar range, oleic acid affects 3-fold stimulation of membrane-preparations of mAC isoform 3 (mAC3) but it does not act on mAC5. Enhanced Gαs-stimulated activities of isoforms 2, 7, and 9, while mAC1 was slightly attenuated, but isoforms 4, 5, 6, and 8 were unaffected. Certain other unsaturated octadecanoic FAs act similarly. FA effects were not observed in AC catalytic domain constructs in which TM domains are not present. Oleic acid also enhances the AC activity of isoproterenol-stimulated HEK293 cells stably transfected with mAC3, although with lower efficacy but much higher potency. Gαs-stimulated mAC1 and 4 cyclase activity were significantly attenuated in the 20-40 micromolar by arachidonic acid, with similar effects in transfected HEK cells, again with higher potency but lower efficacy. While activity mAC5 was not affected by unsaturated FAs, neutral anandamide attenuated Gαs-stimulation of mAC5 and 6 by about 50%. In HEK cells, inhibition by anandamide is low in potency and efficacy. To demonstrate isoform specificity, the authors were able to show that membrane preparations of a domain-swapped AC bearing the catalytic domains of mAC3 and the TM regions of mAC5 are unaffected by oleic acid but inhibited by anandamide. To verify in vivo activity, in mouse brain cortical membranes 20 μM oleic acid enhanced Gαs-stimulated cAMP formation 1.5-fold with an EC50 in the low micromolar range.

Strengths:

(1) A convincing demonstration that certain unsaturated FAs are capable of regulating membrane adenylyl cyclases in an isoform-specific manner, and the demonstration that these act at the AC transmembrane domains.

(2) Confirmation of activity in HEK293 cell models and towards endogenous AC activity in mouse cortical membranes.

(3) Opens up a new direction of research to investigate the physiological significance of FA regulation of mACs and investigate their mechanisms as tonic or regulated enhancers or inhibitors of catalytic activity.

(4) Suggests a novel scheme for the classification of mAC isoforms.

Comments on revised version:

The issues I raised have largely been addressed. A minor concern relates to the legend for Figure 2C, where, according to the author's rebuttal, the vertical axis is "The ratio would be (Gsα + oleic acid stimulation) / (Gsα stimulation)" Otherwise, my general evaluation of the importance of the manuscript stands as stated in my initial review, namely, that the manuscript presents data and results that add a new dimension to existing paradigms for AC regulation, and will prompt future research into the role of physiological lipids in isoform-specific activation or inhibition of AC in tissues.

---

## [Referee Report · Reviewer #3 (Public review)]

Summary:

Landau et al. have submitted a manuscript describing for the first time that mammalian adenylyl cyclases can serve as membrane receptors. They have also identified the respective endogenouse ligands which act via AC membrane linkers to modify and control Gs-stimulated AC activity either towards enhancement or inhibition of ACs which is family and ligand-specific. Overall, they have used classical assays such as adenylyl cyclase and cAMP accumulation assays combined with molecular cloning and mutagenesis to provide exceptionally strong biochemical evidence for the mechanism of the involved pathway regulation.

Strengths:

The authors have gone the whole long classical way from having a hypothesis that ACs could be receptors to a series of MS studies aimed at ligand indentification, to functional studies of how these candidate substances affect the activity of various AC families in intact cells. They have used a large array of techniques with a paper having clear conceptual story and several strong lines of evidence.

Comments on revised version:

In general, the authors have addressed my comments satisfactorily apart from the suggestion to use a lower ISO concentration in their assay or at least to discuss this issue, cite relevant literature etc. Pending this small amendment I would to fine to proceed.

---

## [Author Response]

The following is the authors’ response to the original reviews.

**Public Reviews:**

**Reviewer #1 (Public review):**
Summary:The authors show that the Gαs-stimulated activity of human membrane adenylyl cyclases (mAC) can be enhanced or inhibited by certain unsaturated fatty acids (FA) in an isoform-specific fashion. Thus, with IC50s in the 10-20 micromolar range, oleic acid affects 3-fold stimulation of membrane-preparations of mAC isoform 3 (mAC3) but it does not act on mAC5. Enhanced Gαs-stimulated activities of isoforms 2, 7, and 9, while mAC1 was slightly attenuated, but isoforms 4, 5, 6, and 8 were unaffected. Certain other unsaturated octadecanoic FAs act similarly. FA effects were not observed in AC catalytic domain constructs in which TM domains are not present. Oleic acid also enhances the AC activity of isoproterenol-stimulated HEK293 cells stably transfected with mAC3, although with lower efficacy but much higher potency. Gαs-stimulated mAC1 and 4 cyclase activity were significantly attenuated in the 20-40 micromolar by arachidonic acid, with similar effects in transfected HEK cells, again with higher potency but lower efficacy. While activity mAC5 was not affected by unsaturated FAs, neutral anandamide attenuated Gαs-stimulation of mAC5 and 6 by about 50%. In HEK cells, inhibition by anandamide is low in potency and efficacy. To demonstrate isoform specificity, the authors were able to show that membrane preparations of a domain-swapped AC bearing the catalytic domains of mAC3 and the TM regions of mAC5 are unaffected by oleic acid but inhibited by anandamide. To verify in vivo activity, in mouse brain cortical membranes 20 μM oleic acid enhanced Gαs-stimulated cAMP formation 1.5-fold with an EC50 in the low micromolar range.Strengths:(1) A convincing demonstration that certain unsaturated FAs are capable of regulating membrane adenylyl cyclases in an isoform-specific manner, and the demonstration that these act at the AC transmembrane domains.(2) Confirmation of activity in HEK293 cell models and towards endogenous AC activity in mouse cortical membranes.(3) Opens up a new direction of research to investigate the physiological significance of FA regulation of mACs and investigate their mechanisms as tonic or regulated enhancers or inhibitors of catalytic activity.(4) Suggests a novel scheme for the classification of mAC isoforms.Weaknesses:(1) Important methodological details regarding the treatment of mAC membrane preps with fatty acids are missing.

We will address this issue in more detail.

(2) It is not evident that fatty acid regulators can be considered as "signaling molecules" since it is not clear (at least to this reviewer) how concentrations of free fatty acids in plasma or endocytic membranes are hormonally or otherwise regulated.

Although this question is not the subject of this ms., we will address this question in more detail in the discussion of the revision.

**Reviewer #2 (Public review):**
Summary:The authors extend their earlier findings with bacterial adenylyl cyclases to mammalian enzymes. They show that certain aliphatic lipids activate adenylyl cyclases in the absence of stimulatory G proteins and that lipids can modulate activation by G proteins. Adding lipids to cells expressing specific isoforms of adenylyl cyclases could regulate cAMP production, suggesting that adenylyl cyclases could serve as 'receptors'.Strengths:This is the first report of lipids regulating mammalian adenylyl cyclases directly. The evidence is based on biochemical assays with purified proteins, or in cells expressing specific isoforms of adenylyl cyclases.Weaknesses:It is not clear if the concentrations of lipids used in assays are physiologically relevant. Nor is there evidence to show that the specific lipids that activate or inhibit adenylyl cyclases are present at the concentrations required in cell membranes. Nor is there any evidence to indicate that this method of regulation is seen in cells under relevant stimuli.

Although this question is not the subject of this ms., we will address this question in more detail in the discussion of the revision.

**Reviewer #3 (Public review):**
Summary:Landau et al. have submitted a manuscript describing for the first time that mammalian adenylyl cyclases can serve as membrane receptors. They have also identified the respective endogenouse ligands which act via AC membrane linkers to modify and control Gs-stimulated AC activity either towards enhancement or inhibition of ACs which is family and ligand-specific. Overall, they have used classical assays such as adenylyl cyclase and cAMP accumulation assays combined with molecular cloning and mutagenesis to provide exceptionally strong biochemical evidence for the mechanism of the involved pathway regulation.Strengths:The authors have gone the whole long classical way from having a hypothesis that ACs could be receptors to a series of MS studies aimed at ligand indentification, to functional studies of how these candidate substances affect the activity of various AC families in intact cells. They have used a large array of techniques with a paper having clear conceptual story and several strong lines of evidence.Weaknesses:(1) At the beginning of the results section, the authors say "We have expected lipids as ligands". It is not quite clear why these could not have been other substances. It is because they were expected to bind in the lipophilic membrane anchors? Various lipophilic and hydrophilic ligands are known for GPCR which also have transmembrane domains. Maybe 1-2 additional sentences could be helpful here.

Will be done as suggested.

(2) In stably transfected HEK cells expressing mAC3 or mAC5, they have used only one dose of isoproterenol (2.5 uM) for submaximal AC activation. The reference 28 provided here (PMID: 33208818) did not specifically look at Iso and endogenous beta2 adrenergic receptors expressed in HEK cells. As far as I remember from the old pharmacological literature, this concentration is indeed submaximal in receptor binding assays but regarding AC activity and cAMP generation (which happen after signal amplification with a so-called receptor reserve), lower Iso amounts would be submaximal. When we measure cAMP, these are rather 10 to 100 nM but no more than 1 uM at which concentration response dependencies usually saturate. Have the authors tried lower Iso concentrations to prestimulate intracellular cAMP formation? I am asking this because, with lower Iso prestimulation, the subsequent stimulatory effects of AC ligands could be even greater.

The best way to address this issue is to establish a concentration-response curve for Iso-stimulated cAMP formation using the permanently transfected cells. We note that in the past isoproterenol concentrations used in biochemical or electrophysiological experiments differed substantially.

(3) The authors refer to HEK cell models as "in vivo". I agree that these are intact cells and an important model to start with. It would be very nice to see the effects of the new ligands in other physiologically relevant types of cells, and how they modulate cAMP production under even more physiological conditions. Probably, this is a topic for follow-up studies.

The last sentence is correct.

Appraisal of whether the authors achieved their aims, and whether the results support their conclusions:The authors have achieved their aims to a very high degree, their results do nicely support their conclusions. There is only one point (various classical GPCR concentrations, please see above) that would be beneficial to address.Without any doubt, this is a groundbreaking study that will have profound implications in the field for the next years/decades. Since it is now clear that mammalian adenylyl cyclases are receptors for aliphatic fatty acids and anandamide, this will change our view on the whole signaling pathway and initiate many new studies looking at the biological function and pathophysiological implications of this mechanism. The manuscript is outstanding.
**Recommendations for the authors:**

**Reviewer #1 (Recommendations for the authors):**
It is not clear from the methods section how free FAs were applied to membrane preparations or HEK293 cells. Were FAs solubilized in organic solvents, or introduced as micelles?

The requested info is inserted into the M&M section

Could the authors comment on what is known about the concentration of oleic acid and other non-saturated fatty acids in plasma membranes relative to those required to produce allosteric effects on cyclase activity?

This info is now included in the last paragraph of the discussion.

It would be worthwhile to test the effect of FAs on basal (not Gαs-stimulated) activity of mACs.

This has been carried with mAC isoforms 2, 3, 7, and 9 in which oleic acid enhances Gsα-stimulated activity. Due to the low levels of basal activities interpretable data were not obtained.

Do triglycerides esterified with oleic acid stimulate mAC3 and other sensitive isoforms?

Experiments were done with triolein and 2-oleoyl-glycerol (the answer is no). The data are presented in Fig. 3 and in the appendix Fig.’s 8, 9, 14; structural formulas in appendix 2 Fig. 4 were updated.

Does the quantity plotted on the vertical axis of Figure 1, right panel represent "Fractional Stimulation by Oleic acid" rather than simply "Fold Stimulation"? Clearly, as shown in the two left-most panels, Gαs stimulates both mAC and mAC5. Rather it seems that the ratio (oleic acid stimulation) / (Gαs stimulation) remains constant. This observation supports the statement in the discussion that "We suppose that in mAC3 the equilibrium of two differing ground states favors a Gαs-unresponsive state and the effector oleic acid concentration-dependently shifts this equilibrium to a Gαs-responsive state". It could also be said that the effect of oleic acid is additive, and in constant proportion to that of Gαs.

This comment certainly is related to Fig. 2:

The ratio would be (Gsα + oleic acid stimulation) / (Gsα-stimulation), i.e., fractional stimulation by addition of oleic acid is identical to fold stimulation.

We have amended the legend to fig. 2C for clarification.

The last sentence is wrong because oleic acid alone does not stimulate.

It is stated on page 3, 2nd to last line that "The action of oleic acid on mAC3 was instantaneous...". Since the earliest time point is taken at 5 minutes, the claim that the action of the lipid is instantaneous cannot be made. Information about kinetics would be useful to have, since it is possible that the lipid must be released from a micelle and be incorporated into the AC membrane fraction before it is active.

The first point is 3 min.

We deleted the word “instantaneous” and added the correlation coefficients for both conditions in the legend to appendix 2; fig. 1 for clarification.

The data spread in Figure 4 and other figures showing similar data is significant, to the extent that the computed value for EC50 may not be of high precision. Authors should cite the correlation coefficient for the overall fit and uncertainty for the EC50 value (in addition to significances by t-test of individual data points).

This will not add valuable information. Pearsons correlation coefficients are only for linear relationships.

(cf. N.N. Kachouie, W. Deebani (2020) Association Factor for Identifying Linear and Nonlinear Correlations in Noisy Conditions. Entropy 22:440)

The "switch" between relatively low potency and high efficacy in membrane preps to high potency and low efficacy in cells is remarkable. Could this have a methodological basis or is it reflective of the mechanism by which FAs access mACs in membrane preps vs. cell membranes, or perhaps some biochemical transformation of the lipid in cells?

Honestly, we do not know.

The authors should note that there is some precedence for this work:J Nakamura , N Okamura, S Usuki, S Bannai, Inhibition of adenylyl cyclase activity in brain membrane fractions by arachidonic acid and related unsaturated fatty acids. Arch Biochem Biophys. 2001 May 1;389(1):68-76. doi: 10.1006/abbi.2001.2315.The effects of FA deficiencies on AC and related activities have been noted:Alam SQ, Mannino SJ, Alam BS, McDonough K Effect of essential fatty acid deficiency on forskolin binding sites, adenylate cyclase, and cyclic AMP-dependent protein kinase activity, the levels of G proteins and ventricular function in rat heart. J Mol Cell Cardiol. 1995 Aug;27(8):1593-604. doi: 10.1016/s0022-2828(95)90491-3. PMID: 8523422The latter publications are supportive of, and provide context to, the author's findings.

Both references are mentioned and cited.

Minor points:The significance of the coloring scheme in Figure 5C bar graph should be stated in the legend.

Done.

In the introduction, it is stated that "The protein displayed two similar catalytic domains (C1 and C2) and two dissimilar hexahelical membrane anchors (TM1 and TM2)". In both cases, the respective domains can be said to be similar in overall fold, but - certainly in the case of the catalytic domains - different in amino acid sequence in functionally important regions of the domain.

Done: Changed wording.

The statement in the introduction that "The domain architecture, TM1-C1-TM2-C2, clearly indicated a pseudoheterodimeric protein composed of two concatenated bacterial precursor proteins" The authors refer to the fact that mammalian enzymes are pseudo heterodimers whereas bacterial type III cyclases are dimers of identical subunits.

Done.

**Reviewer #2 (Recommendations for the authors):**
The title need not state that a 'new class of receptors' has been identified. There is no direct evidence that the lipids bind to the enzymes, and the affinities can only be surmised from the EC50 graphs. To call a protein a receptor requires evidence to show that the binding is specific by showing that binding can be inhibited by a large excess of 'unlabelled' ligand. This could have been done by procuring labelled lipids for experimental verification.

As is well known, lipids easily bind to proteins. In this study no purified proteins were used. Therefore, binding assays most likely would result in unreliable data.

The paper would have benefitted from showing sequence alignments in the TM domains of the ACs discussed in the paper. Further, a phylogenetic tree of mammalian ACs would also reveal which enzymes from other species may be regulated similarly to those described in the paper. This would be important for researchers who use other model organisms to study cAMP signalling.

Such data are in multiple papers accessible in the literature. Where deemed appropriate we inserted references.

Figures 1A and 1B show data from only two experiments. A third experiment would have been useful in order to show the statistical significance of the data.

At this stage more experiments would not have affected further experimental plans.

Statements made in the text (for example, the last paragraph on page 6) state only the mean value and not the SDs. This would have been important to include even if the data is shown in the appendix. The same is true in the Legend of Figure 2. Why have the authors decided to use SEM and not SDs?

The reason is specified in M&M.

Concentrations of lipids used in biochemical assays are in the micromolar range. This suggests that we have moderate affinity binding, more in the range of an enzyme for a substrate rather than a receptor-ligand interaction.

We happen to disagree. Clearly, the differential activities, enhancing or attenuating Gsα-stimulated mAC activities is most plausibly explained by mAC receptor properties. mACs have enzyme activities using fatty acids as substrates.

The authors add lipids to cells and show changes in cAMP levels in their presence and absence. They also discuss how these extracellular lipids could be produced. Do you think this is necessary in vivo, though? Could the lipids present in membranes naturally act as regulators? Do specific lipid concentrations differ in different cell types, suggesting tissue-specific regulation of these mammalian Acs?These are things that could be discussed in the manuscript.

The last paragraph of the discussion deals with these questions.